

# Optimization of Aeolus Optical Properties Products by Maximum-Likelihood Estimation

Frithjof Ehlers[1], Thomas Flament[2], Alain Dabas[2], Dimitri Trapon[2], Adrien Lacour[2], Holger Baars[3], and Anne Grete Straume-Lindner[1]

[1]ESA-ESTEC, Keplerlaan 1, 2201 AZ Noordwijk, Netherlands
[2]Météo-France, CNRS, Toulouse, France
[3]Leibniz-Institut für Troposphärenforschung e.V., Permoserstraße 15, 04318 Leipzig, Germany

**Correspondence:** Frithjof Ehlers (frithjof.blablub@gmail.com)

**Abstract.** The European Space Agency (ESA) Earth Explorer Mission, Aeolus, was launched in August 2018 and embarks the first Doppler Wind Lidar in space. Its primary payload, the Aeolus LAser Doppler INstrument (Aladin) is a Ultra Violet (UV) High Spectral Resolution Lidar (HSRL) measuring atmospheric backscatter from air molecules and particles in two separate channels. The primary mission product is globally distributed line-of-sight wind profile observations in the troposphere and lower stratosphere. Atmospheric optical properties are provided as a spin-off product. Being and HSRL, Aeolus is able to independently measure the particle extinction coefficients, co-polarized particle backscatter coefficients and the co-polarized lidar ratio. This way, the retrieval is independent of a-priori information. The optical properties are retrieved using the Standard Correct Algorithm (SCA), which is an algebraic inversion scheme to a (partly) ill-posed problem and therefore sensitive to measurement noise. In this work, we rephrase the SCA into a physically constrained Maximum Likelihood Estimation (MLE) problem and demonstrate predominantly positive impact and considerable noise suppression capabilities. These improvements originate from the use of all available information within the SCA in conjunction with the expected physical bounds concerning the expected range of the lidar ratio. The new MLE algorithm is equally evaluated against the SCA on end-to-end simulations of two homogeneous scenes and for real Aelous data collocated with measurements by a ground-based lidar and the CALIPSO satellite to consolidate and to illustrate the improvements. The largest improvements were seen in the retrieval of the extinction coefficients and lidar ratio ranging up to one order of magnitude or more in some cases due to an effective noise dampening.

## 1 Introduction

Aeolus is an ESA (European Space Agency) Earth Explorer Core Mission launched on 22 August 2018 (Stoffelen et al., 2005; ESA, 2008). Aeolus' payload consists of the Atmospheric LAser Doppler INstrument (ALADIN), which is a UV high spectral resolution (HSRL) Doppler Wind Lidar operating at 355 nm wavelength (Chanin et al., 1989; Garnier and Chanin, 1992; Korb et al., 1992; Souprayen et al., 1999b,a; Gentry et al., 2000) and the first Doppler Wind Lidar in space. The primary mission goal is to provide accurate global measurements of vertical wind profiles in the troposphere and lower stratosphere with global coverage each week for use in operational Numerical Weather Prediction (NWP) and scientific research. Additionally, Aeolus can contribute to the global monitoring of cloud and aerosol optical properties due to the applied aerosol HSRL method



(Shipley et al., 1983; Shimizu et al., 1983; Grund and Eloranta, 1991; She et al., 1992; Weitkamp, 2006). Among various space
missions that carry active lidar instruments are the Cloud and Aerosol Lidar and Infrared Pathfinder Satellite Observations
(CALIPSO) launched in 2006 (Winker et al., 2003), Ice, Cloud, and land Elevation Satellites (ICESat and ICESat-2), launched
in 2003 and 2018 (Spinhirne et al., 2005; Martino et al., 2019), respectively, the Cloud-Aerosol Transport System (CATS)
deployed on the international space station (ISS) in 2015 (McGill et al., 2015). A further mission currently being implemented
is ESA's Earth Clouds, Aerosols, and Radiation Explorer (EarthCARE) scheduled for launch in 2022 (Illingworth et al., 2015).
Particularly the combination of CALIPSO, CATS, Aeolus and EarthCARE will potentially offer a detailed and long term
dataset of aerosol and cloud (optical) properties to the benefit of numerical weather prediction and climate (change) research,
because high uncertainties in climate change modelling regard the indirect effect of aerosols on clouds and anthropogenic ra-
diative forcing (Illingworth et al., 2015). This dataset is a unique addition to ground based lidar networks such as the European
Aerosol Research Lidar Network (EARLINET) (Pappalardo et al., 2014) due to the aspect of regular global coverage. The
key advantage of ALADIN's HSRL-capability is the independent estimation of volume extinction coefficient and co-polarized
volume backscatter coefficient products at 355nm from two different spectral channels. On the other hand, this requires a ro-
bust channel cross-talk correction. The Aeolus atmosheric optical properties retrieval is implemented in the so-called Level
2A processor, as described by Flamant et al. (2008) and Flamant et al. (2017). Following its launch, the Aeolus atmospheric
backscatter signal levels were found to be a factor of 2.5 to 3 lower than expected due to lower laser output energies and a
decreased instrument transmission by about 30% (e.g. Reitebuch et al. (2020)). This has caused lower signal-to-noise ratios
(SNR), and as a result, Aeolus optical properties retrieval with the HSRL standard correct algorithm (SCA) is hampered due
to high noise sensitivity, see Appendix A. Particularly the particle extinction coefficient retrieval is severely affected, due to its
dependency on the slope of already noisy attenuated molecular backscatter signals. In the past, attempts were made to mitigate
nonphysical optical properties in SCA (such as oscillating/negative extinction coefficients in low aerosol load conditions) by
measures like zero-flooring or signal accumulation in even coarser range bins (Flamant et al., 2017) but with limited success.
Particle extinction coefficient retrieval from HSRL and similarly Raman lidar observations is known to be an ill-posed problem
in presence of any noise (Shcherbakov, 2007; Pornsawad et al., 2008, 2012; Denevi, 2015; Garbarino et al., 2016). A classical
mitigation approach is to increase SNR by averaging the data in non-overlapping blocks before processing or application of
low-pass filters either on the measured lidar signal or the atmospheric optical properties, i.e., aerosol backscatter and extinction
coefficients (Ansmann et al., 2007; Young et al., 2008; Eloranta, 2014; Flamant et al., 2017). Here, the lidar signal is seen as a
two dimensional image with dimensions range and time owing to continuous operation. But decreased resolution is often not
acceptable due to increase of representativeness errors, e.g., when the heterogeneity of the observed atmospheric scene forbids
a coarser description in case of high gradients (broken clouds with aerosols). There is no suitable resolution in between (i) low
SNR / noise dominated regime and (ii) representativeness error dominated regime, so that suitable regularization techniques/
non-linear regression methods must be applied. The most commonly used methods for retrieval of atmospheric optical prop-
erties from active optical remote sensing by lidars are (penalized) Least Square Fit (LSF) (Whiteman, 1999; Pornsawad et al.,
2008, 2012), (Penalized) Maximum Likelihood Estimation (PMLE) (Shcherbakov, 2007; Denevi, 2015; Garbarino et al., 2016;
Marais et al., 2016; Xiao et al., 2020) and Optimal Estimation Method or Bayesian method (OEM) (Povey et al., 2014; Sica and



Haefele, 2015; Donovan, D.P. et al., 2020). A thorough documentation of OEM in inverse problems for atmospheric sounding was given by Rodgers (2000), whose notation will be adapted in this work for all non-linear regression methods. The strengths of such techniques lie within the characterization and utilization of any additional information, such as the measurement uncertainties or another hypothesis about the state. Such additional information content (if correct) enables a better characterization of the underlying aerosol optical properties. Most of the mentioned works exploit the knowledge of measurement uncertainties and positivity constraints on optical properties and can therefore outperform purely algebraic inversions of (particle) extinction coefficients. The SCA approach is such a purely algebraic inversion algorithm. Another specific advantage in the works of Shcherbakov (2007); Povey et al. (2014); Marais et al. (2016); Xiao et al. (2020) is the coupled retrieval of particle backscatter coefficients and extinction coefficients via the particle lidar ratio (extinction-to-backscatter ratio), because particle backscatter coefficients are usually known with much higher precision and at finer resolution. Thus, the particle exctinction can be well located along the profile and may vary only in terms of the typical lidar ratio range. This way, the retrieved set of optical properties are automatically consistent in itself and with the underlying physics (assumptions).

In this work, we want to explore and demonstrate the potential of non-linear regression in Aeolus optical properties retrieval. The already implemented SCA approaches will serve as a benchmark for comparison. Therefore, the retrieval problem is rephrased into a Maximum Likelihood Estimation (MLE) problem, which aims to solve the noted issues in the SCA algorithms as follows: Firstly, we account for the noise of the signals in both channels and, secondly, suggest that particle backscatter and extinction coefficients are retrieved in a coupled way. This means that the lidar ratio will be constrained to values in between 2 sr and 200 sr. Additionally, a positivity constraint on the extinction and backscatter is set. A considerable gain in the quality of the retrievals is expected, because a coupled retrieval in conjunction with a box-constrained set of space variables will enable the processor to automatically detect and suppress some dominant, anti-correlated noise that originates from the channel crosstalk correction (Appendix A). Therefore, in contrast to the approaches in Shcherbakov (2007); Marais et al. (2016); Xiao et al. (2020), which first obtain backscatter coefficients and successively calculate extinction coefficients, we retrieve both simultaneously. The developed MLE framework naturally offers the potential to be further refined into an Optimal Estimation Method (OEM) or Penalized Maximum Likelihood Estimator (PMLE), due to the strong similarity between the methods.

This paper is structured as follows: A brief instrument description in Sect. 2 is followed by a recap of the classical retrieval algorithms (SCA) and their underlying set of equations and the MLE approach in Sect. 3. In Sect. 4, results are presented and discussed from the comparisons of the SCA and MLE approaches using end-to-end simulations on homogeneous standard atmosphere scenes, real Aeolus observations of a Saharan Air Layer event and Aeolus observations collocated with ground based lidar observations near Tel-Aviv. The final Sect. provides the study conclusions and out-look.

## 2 Instrument

Aeolus revolves the earth in a Sun-synchronous polar orbit at about 320 km altitude with a seven days repeat cycle. The ALADIN instrument emits a narrow-bandwidth UV laser pulse close to 355 nm wavelength and the instrument is pointing to



earth with off-nadir slant angle of 35° measured from the spacecraft, which accounts for approximately $(37 \pm 0.2)°$ off-nadir angle at the Earth surface due to earth's curvature. The diameter of the laser footprint is about 12-15m and the instrument field of view $19\mu$rad. The laser light is backscattered by means of particles (aerosol and hydrometeors resulting in spectrally narrow Mie scattering) and air molecules (resulting in thermal and pressure broadened Rayleigh Brillouin scattering). The frequency of the atmospheric backscattered laser light is Doppler-shifted relative to the emit frequency owing to the relative velocity of the scattering media (wind speed) along the instrument line-of-sight (LOS). The contributions of Earth's rotation and satellite movement are compensated by Aeolus' attitude control and on-ground data processing, such that the Doppler shift is entirely dominated by the LOS wind speeds. In order to measure the LOS wind speeds, Aeolus utilizes two different receiver spectrometers, namely a fringe imaging Fizeau interferometer for the spectrally narrow Mie back-scattered laser light, and two sequential, double edge Fabry-Pérot interferometers for the spectrally broad Rayleigh back-scatter. Hence, continuous LOS wind speed profiles up to 30 km altitude can be measured regardless of the presence and absence of aerosol, unless optically thick features such as dense liquid water clouds block the beam. Additionally, the measured signal intensities allow a retrieval of particle optical properties.

The core of ALADIN is its diode-pumped frequency tripled Nd:YAG laser with 80mJ nominal pulse emit energy and 50Hz pulse repetition frequency. The emit beam is circularly polarized, but the cross-polarized part of the backscattered light is discarded in the receive path optics due to the instrument design. Hence, in the case of strongly depolarizing targets, the signal measured at the detectors is strongly reduced with respect to non-depolarizing targets (ESA, 2008; Flamant et al., 2017). Additionally, the expected atmospheric return signal in orbit is a factor of 2.5 to 3 lower than expected before launch, due to lower laser output energies than originally intended (45 - 72 mJ) and decreased instrument transmission by about 30%, which has caused a lower SNR since mission start (Reitebuch et al., 2020).

The light backscattered from the atmosphere is an attenuation of Mie and Rayleigh backscatter, which is then separated by the two instrument receivers. Figure 2 in Ansmann et al. (2007) illustrates how the spectral characteristics of the Mie and Rayleigh backscatter are exploited to seperate them with the spectrometers (channels) (Ansmann et al., 2007; Reitebuch et al., 2018a). If the LOS wind speed is non-zero, the whole spectrum will be shifted relative to the channel transmission curves due to the Doppler shift. The light first travels to the Mie receiver, where the frequency narrow Mie peak (fringe) is measured to determine this Doppler shift directly. The frequency broadened Rayleigh scattered light gets reflected on the Mie channel and is then directed to the two Rayleigh channel filters. These capture different signal intensities in the two filters centred on each side of the centre emit frequency. By comparing and normalizing the responses in the two filters, the LOS wind Doppler shifts can be calculated. However, the return signals from particles (Mie) and molecules (Rayleigh) are not entirely seperated due to the overlap of the transmission functions of the two channels and the sequential design of the receivers. Therefore, channel crosstalk is present, which needs to be corrected for to obtain un-biased LOS wind and optical properties.

The atmospheric echoes from the single pulses passing through the instrument receivers are collected with time gated accumulation charge coupled devices (ACCDs). The time it takes for the light to travel from the instrument, through the atmosphere and back to the receivers, is used to accumulate the light from the individual pulses over time equivalent to atmospheric vertical bins of 250 m. The same ACCDs, with a quantum efficiency of 85%, are used for both the particle (Mie) and molecular



(Rayleigh) channels. In order to achieve sufficient charge build-up before read-out and digitalization, 19 laser pulses are accumulated directly on the storage columns of the ACCD. The accumulation of 19 laser pulses corresponds to an on-ground distance of about 2.9km along track and is the smallest horizontal measurement which is down-linked to Earth. The vertical
resolution of the detectors can be independently varied between 250m and 2000m in steps of 250m with a total number of ACCD rows and hence vertical range bins of 25. Of these range bins, one is used for solar background measurements, and one is used to sample the ground. In order to achieve the mission requirements for wind random errors, a number of measurements is added up onto coarser scale during the on-ground processing. For Rayleigh winds, a total of 30 measurements are integrated to one so-called Basic Repeat Cycle (BRC) or Observation, equivalent to approximately 87 km along track distance on ground.
Mie winds can be provided at smaller scales dependend on SNR of the aerosol feature. High SNR is also critical for optical properties retrieval, particularly particle extinction coefficients. Hence, optical properties are primarily evaluated on observation scale as well and only refined afterwards in the so called group product. An example of raw signals on measurement scale (2.9 km) and accumulated onto observation scale (87 km) is given in Figure 1, which shows the test case discussed in Section 4.3.
140       The geolocated measurement data from the satellite, including detected Doppler shifts and the so called useful signals from the two Rayleigh and the Mie channel, are provided in the level 1B (L1B) data product (Reitebuch et al., 2018a). These are the signals that have been corrected for the detection chain offset (DCO, measured for all range bins in seperate, non-illuminated pixels), dark current charge offset in memory zone (DC or DCMZ, from on-ground characterization), tripod obscuration (TOBS, in Mie channel only) and the solar background contribution (measured in range bin 25 for all channels)
(Reitebuch et al., 2018a). The L1B product in combination with additional calibration data from the so-called CAL-Suite processing step and meteorological information from a global weather forecast from the European Centre of Medium-Range Weather Forecasts (ECMWF), is used as input to the optical properties processor to generate the Level 2A (L2A) data product (Flamant et al., 2017). The calibration data is obtained from designated instrument modes and internal reference measurements (Reitebuch et al., 2018a). Within the L2A product, three different algorithms are implemented: The first is the so-called Stan-
dard Correct Algorithm (SCA), which makes use of the full HSRL potential and uses both channels to retrieve lidar ratios directly from the data. The second is the Mie channel algorithm (MCA), which applies a Klett-like retrieval (Klett, 1981; Fernald, 1984) with an a-priori lidar ratio value (extinction-to-backscatter ratio). The third, the Iterative Correct Algorithm (ICA), intends to refine the SCA results on finer vertical scales, but fails to generate reasonable results in the nominal operation due to pronounced sensitivity to noise. Hence, given its more accurate approach and better performance, only the SCA algorithm
is considered in the remainder of this paper.

## 3   Methods

The atmospheric forward model in lidar applications is based on the lidar equation (Weitkamp, 2006). In its simplest form, the lidar equation reads

$$s(r) = KG(r)\beta(r)T(r)^2. \tag{1}$$



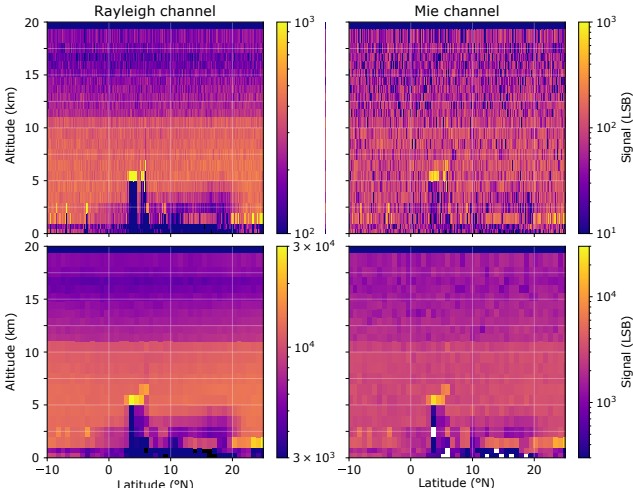

**Figure 1.** Exemplary, raw, in-orbit Rayleigh and Mie 'useful signals' from the L1B product in least significant bits (LSB) on measurement scale (top) and after accumulation on observation scale (bottom) showing the input to the test case that will be discussed in section 4.3.

The signal power $s$ received from distance $r$ is made up of four factors. The constant $K$ summarizes the signal transmission through the lidar instrument, and $G(r) = O(r)r^{-2}$ contains all range dependent terms regarding the measurement geometry. The two unknown terms that contain information on the atmospheric state are the total backscatter coefficient $\beta(r)$ at distance $r$ and the atmospheric transmission $0 < T(r) < 1$ that describes how much light gets lost on the way from the lidar to the target at a distance $r$. In the following we consider the case of the single scattering approximation, i.e., multiple scattering effects

are neglected. As discussed in Flamant et al. (2008) and Ansmann et al. (2007), this is a valid assumption due to the small divergence and narrow FOV of the ALADIN instrument. In case of ALADIN, the lidar equations for the two channels read

$$s_{\mathrm{ray}}(r) = \frac{K_{\mathrm{ray}}N_p E_0}{r^2}\Big[\beta_m(r)C_1(p,t,f) + \beta_{||,p}(r)C_2(f)\Big]T_m^2(r)T_p^2(r) \qquad (2)$$

$$s_{\mathrm{mie}}(r) = \frac{K_{\mathrm{mie}}N_p E_0}{r^2}\Big[\beta_m(r)C_4(p,t,f) + \beta_{||,p}(r)C_3(f)\Big]T_m^2(r)T_p^2(r) \qquad (3)$$

with the Mie and Rayleigh channel signals $s$, laser pulse energy $E_0$, number of accumulated pulses $N_p$, atmospheric temper-

ature $t$, atmospheric pressure $p$, Doppler shift $f$, instrumental calibration constants $K_{\mathrm{ray}}$ and $K_{\mathrm{mie}}$ and crosstalk coefficients $C_{1...4}$ accounting for the fractions of molecular and particulate signal in the Rayleigh and Mie channel, respectively. The total backscatter coefficient has been split into molecular contribution and particle contribution $\beta = \beta_m + \beta_p$. Here, $\beta_p$ is explicitly split into the cross-polarized and co-polarized fraction $\beta_p = \beta_{\perp,p} + \beta_{||,p}$, of which only the co-polarized particle backscatter coefficient is measured due to instrument design. $T_{\mathrm{label}}^2$ denotes the two way transmission

$$T_{\mathrm{label}}^2(r) = \exp\left(-2\int_0^r \alpha_{\mathrm{label}}(u)du\right) \qquad (4)$$

with range dependent extinction coefficient $\alpha$ and label $m$ for molecules and $p$ for particles, respectively. The two unknown parameters of interest are co-polarized particle backscatter coefficient $\beta_{||,p}$ and particle extinction coefficient $\alpha_p$, which can





in principle be solved with the two equations because $C_1 > C_2 > 0$ and $C_3 > C_4 > 0$ holds true by instrument design. In the following, we are also making excessive use of the co-polarized lidar ratio (extinction-to-backscatter ratio) $\gamma_{||,p,i} = \alpha_{p,i}/\beta_{||,p,i}$.

Since the co-polarized particulate backscatter $\beta_{||,p}$ is lower than the total particulate backscatter $\beta_p$, the co-polarized lidar ratio overestimates the true lidar ratio ($\gamma_{||,p} > \gamma_p = \alpha_p/\beta_p$). The lidar equations can be lightened by introduction of the range resolved atmospheric signals at telescope entry

$$X(r) = \frac{\beta_m(r)}{r^2} T_m(r)^2 T_p(r)^2 \tag{5}$$

$$Y(r) = \frac{\beta_{||,p}(r)}{r^2} T_m(r)^2 T_p(r)^2 \tag{6}$$

with $X(r)$ for molecular backscatter and $Y(r)$ for particulate backscatter in units $\mathrm{m}^{-3}\mathrm{sr}^{-1}$. These resemble normalized signals that would be obtained in the absence of channel crosstalk. So the lidar equations read

$$s_{\mathrm{ray}} = K_{\mathrm{ray}} N_p E_0 \Big[ C_1 X + C_2 Y \Big] \tag{7}$$

$$s_{\mathrm{mie}} = K_{\mathrm{mie}} N_p E_0 \Big[ C_4 X + C_3 Y \Big] \tag{8}$$

when variable dependences are dropped for the sake of readability.

## 3.1 Standard Correct Algorithm (SCA)

For a more detailed description of the SCA and a discussion of its short comings, we refer to Appendix A and Flamant et al. (2017). In the following, the index $i \le n = 24$ as subscript to the properties above denotes the range bin index. This implies for the signals $s$, $X$ and $Y$ that the property has been integrated over a discrete range $[R_{i-1}, R_i]$, i.e., $s_{\mathrm{ray},i} = \int_{R_{i-1}}^{R_i} s_{\mathrm{ray}}(r)dr$. For all other variables like backscatter coefficients $\beta$, extinction coeffcients $\alpha$ and range $R$ this subscript denotes the average

in range bin $i$, i.e., $\beta_{||,p,i} = \frac{1}{\Delta R_i} \int_{R_{i-1}}^{R_i} \beta_{||,p}(r)dr$ with $\Delta R_i = R_i - R_{i-1}$ and equivalently for subscript $m$. As a consequence, particle optical depth of a bin is denoted $L_{p,i} = \alpha_{p,i} \Delta R_i$. The following approximations for the range corrected signals are made by using the mean bin properties from above (Flamant et al., 2017):

$$X_i \approx \frac{\Delta R_i T_{m,i}^2 \beta_{m,i}}{R_i^2} e^{-L_{m,i}} \left( \frac{1 - e^{-2L_{p,i}}}{-2L_{p,i}} \right) \cdots$$
$$\cdot \exp\left( -2L_{p,sat} - 2\sum_{j=0}^{i-1} L_{p,j} \right) \tag{9}$$

$$Y_i \approx \frac{T_{m,i}^2}{R_i^2} e^{-L_{m,i}} \left( \frac{1 - e^{-2L_{p,i}}}{-2\gamma_{||,p,i}} \right) \cdots$$
$$\cdot \exp\left( -2L_{p,sat} - 2\sum_{j=0}^{i-1} L_{p,j} \right). \tag{10}$$

with unknown optical depth $L_{p,sat}$ in between telescope and first range bin. With this and equations (7) and (8), SCA solves algebraically for the two unknowns, co-polarized lidar ratios $\gamma_{||,p,i}$ and optical depths $L_{p,i}$ (and $L_{p,sat}$), which can be rephrased into / are equivalent to extinction coefficients $\alpha_{p,i}$ and backscatter coefficients $\beta_{||,p,i}$ (and $L_{p,sat}$). Backscatter coefficients are





simply retrieved from $\beta_{||,p,i} = Y_i \beta_{m,i}/X_i$. The SCA algorithm produces two sets of products, the SCA and the so-called SCA

midbin backscatter and extinction coefficient profile products and lidar ratios. The SCA midbin product is averaging the SCA

neighbouring bins onto a coarser resolution in order to dampen oscillations in the retrieved profiles in scenes with low signals

in order to obtain a more stable product. Further details of the products and their performances is provided in section 4 and

annex A.

## 3.2   Maximum Likelihood Estimation (MLE) Retrieval

The basis of MLE and OEM methods is formed by the forward model $\boldsymbol{y} = F(\boldsymbol{x})$, that maps physical properties from state space

on measurement space (Rodgers, 2000). We use equations (7) to (10) in order to calculate the measurement space variables

(signals) from the state space variables (optical properties) with

$$
\boldsymbol{y} = \begin{pmatrix} s_{\mathrm{ray},0} \\ \vdots \\ s_{\mathrm{ray},n} \\ s_{\mathrm{mie},0} \\ \vdots \\ s_{\mathrm{mie},n} \end{pmatrix} \quad \text{and} \quad \boldsymbol{x} = \begin{pmatrix} L_{p,0} \\ \vdots \\ L_{p,n} \\ \gamma_{||,p,0} \\ \vdots \\ \gamma_{||,p,n} \\ L_{p,sat} \end{pmatrix}, \quad n \leq 24.
\tag{11}
$$

The deviations between the actual measurement and the forward modelled state are comprised in a cost function that needs

to be minimized to obtain a good estimate of the underlying true state. Generally, the cost function in non-linear regression

problems is composed of two terms

$$
J(\boldsymbol{x},\boldsymbol{y}) = J_{\mathrm{obs}}(\boldsymbol{x},\boldsymbol{y}) + J_{\mathrm{prior/constraint}}(\boldsymbol{x})
\tag{12}
$$

of which the first describes deviation from the measurement and the second the deviation from some *a-priori* state (applied in

OEM) or from another *a-priori* constraint (applied in PMLE), e.g., smoothness. It is also referred to as the penalty term. In

MLE, $J$ is the log-likelihood function, which in the case of normally distributed measurement errors becomes the weighted

least squares term

$$
J = [\boldsymbol{y} - \boldsymbol{F}(\boldsymbol{x})]^{\mathsf{T}} \mathbf{S_y} [\boldsymbol{y} - \boldsymbol{F}(\boldsymbol{x})]
\tag{13}
$$

with measurement error covariance matrix $\mathbf{S_y}$. Often the choice of Poisson noise in conjunction with the Kullback-Leibler

Divergence is preferred in Lidar applications (Denevi, 2015; Marais et al., 2016; Garbarino et al., 2016; Weitkamp, 2006),

because photon shot-noise is the dominant noise source for the signals on the detectors, which is fairly Poisson distributed.

Here, we do not restrict noise amplitudes to the Poisson case to account for additional noise contributions, such as laser pulse

frequency jitter, ACCD readout noise, dark electron/thermal noise contribution and potentially unknown 'noise' sources such

as atmospheric variability. The description of $J_{\mathrm{obs}}$ in terms of normal distributions is not a critical aspect of the method, because





the Poisson noise distribution becomes indistinguishable from a Gaussian given sufficient signal accumulation (central limit theorem). For the time being, no a-priori term/constraint contributes to the cost function used throughout this work, although limits will be imposed on the state space variables by solving specifically the box-constrained MLE problem

$$\min_{\substack{\boldsymbol{x};\\ 2\,\mathrm{sr}<\gamma_{||,p}<200\,\mathrm{sr};\ 0<L_p}} [\boldsymbol{y}-\boldsymbol{F}(\boldsymbol{x})]^{\mathsf{T}}\,\mathbf{S_y}\,[\boldsymbol{y}-\boldsymbol{F}(\boldsymbol{x})] \tag{14}$$

with box-constraints on lidar ratio and optical depth. The state that solves this minimization problem (14) is denoted $\boldsymbol{x}^*$.

In practice, the minimization problem (14) needs to be solved for all optical properties profiles along Aeolus orbit. So instead of solving (14) per single atmospheric column, it is solved for all columns at once (on observation scale). Without loss of generality, this can be realized by minimisation of the sum of the single cost functions

$$\min_{\substack{\boldsymbol{x}_1,\,...,\,\boldsymbol{x}_N;\\ 2\,\mathrm{sr}<\gamma_{||,p}<200\,\mathrm{sr};\ 0<L_p}} \sum_{k}^{N} J_k(\boldsymbol{y}_k,\boldsymbol{x}_k) \tag{15}$$

over all $N$ lidar profiles with index $k$. This is equivalent to the ensemble of the seperate minimization problems because the
$k$-th cost function is strictly positive and only sensitive to changes of the $k$-th state vector. So the collection of state vectors that minimizes the sum of the cost functions has to minimize each summand independently. This way, we gain the freedom to use the inherent 2D information of the lidar signal in future developments, e.g., to couple neighboring profiles by the introduction of a regularization term to (15) that acts on the horizontal direction along the satellite orbit (Marais et al., 2016; Xiao et al., 2020).


     Principally, extinction and backscatter coefficients $(\alpha_p,\beta_{||,p})$ can be chosen as state space description, as well as lidar ratio and backscatter $(\gamma_{||,p},\beta_{||,p})$. Though, the reason to favor a state space description containing the lidar ratio is that such states can be easily constrained to physical bounds by forcing upper and lower ranges within the retrieval. Within the $(\alpha_p,\beta_{||,p})$-description, the lidar ratio constraint would become non-linear and harder to handle by off-the-shelf tools for numerical opti-
mization. An equally valid choice of the measurement space variables $\boldsymbol{y}$ are the crosstalk-corrected signals $X_i$ and $Y_i$ instead of the Rayleigh and Mie channel signals $s_{\mathrm{ray},i}$ and $s_{\mathrm{mie},i}$, but then the measurement covariance matrix $\mathbf{S_y}$ would show off diagonal entries due to the linear transform in (A4-A5). The above choice has been made for the sake of simplicity and to lighten the cost function and its gradients, since the signals in seperated vertical range bins are expected to be uncorrelated except for a small range bin overlap (Weiler, 2015). This accounts for $\pm120$m altitude for Mie ACCD and $\pm30$m for Rayleigh ACCD in
nominal operation and regardless of range bin settings. As in SCA, this overlap will not be considered in the following.

     As pointed out by Povey et al. (2014), unbiased uncertainty estimates are a prerequisite to obtain good results. This is an issue for low signal intensities when estimating the Poisson uncertainty from the uncertain signal itself, i.e., $\hat{\sigma}_s=\sqrt{s+\varepsilon_s}$ with proxy for standard deviation $\hat{\sigma}_s$, true signal $s$ and actual Poisson noise value $\varepsilon_s$: The uncertainty will be biased by the exact
noise value in the signal. In Aeolus L1B data products, the Poisson noise assumption is applied to calculate signal-to-noise ratio (SNR) in both signal channels, including the solar background contribution. But here, we use instead the scaled variance





of the signals on measurement scale, at 2.9 km, to approximate the noise level in the bins on observation scale, at 87 km, under the assumption of a homogeneous scene, see Appendix B.

The choice of the upper and lower lidar ratio bounds takes the following points into consideration: On one hand, the true lidar ratio at 355nm is expected to exceed values of 100 sr only in rare cases and a physically lower bound might be presented by approximately 10 sr, see Fig. 8 in Illingworth et al. (2015) or (Wandinger et al., 2015). On the other hand, the coarse vertical resolution and the effects of depolarization need to be accounted for. Therefore, highly depolarizing aerosol, such as desert dust and ice particles in cirrus clouds, will appear highly attenuating, since the co-polarized lidar ratio can be artificially increased

by a factor up to $1.85$ for desert dust and 3 for cirrus clouds compared to the true lidar ratio. Taking into account typical lidar ratio to depolarization distributions as in Wandinger et al. (2015); Illingworth et al. (2015), values as high as 130 sr will be well within physical limits for Aeolus. Additionally, as shown in Flamant et al. (2017), different hypothesis on the distribution of aerosol layers within a range bin can easily alter the true particle optical depth retrieval results by a factor of 16. As the same systematic errors apply to co-polarized particle backscatter coefficients, less variability is expected for the co-polarized particle

lidar ratio estimates. However, the applied bounds need to account for the forward model errors by an extra margin. All things considered, the limits of 2 sr to 200 sr resemble a reasonable trade-off.

    The minimization problem (15) is solved approximately by the L-BFGS-B algorithm, a limited-memory quasi-Newton code for bound-constrained optimization. More precisely, the implementation described in Zhu et al. (1997), version 3.0, is used.

As an addition, the cost function gradient is evaluated efficiently via automatic differentiation. The initial conditions, or the *first guess*, consist of aerosol free atmosphere with $L_p$=0 and a lidar ratio of $\gamma_{||,p} = 60$ sr. Since the L-BFGS-B algorithm is not invariant under variable transforms, it was necessary to introduce an additional scale parameter in the state vector for good convergence rates of the cost function. The applied transformation maps $L_p \rightarrow 200L_p$ in the state vector to ensure that all entries are about the same order of magnitude. More advanced variable transforms such as pre-whitening of the variables

(Rodgers, 2000) may be appropriate to optimise performance, but the proposed rescaling is found to be sufficient.

MLE estimates usually suffer from overfitting and noise amplification. Therefore, an implicit regularization is often achieved by optimal choice of the number of iterations (Denevi, 2015; Garbarino et al., 2016). But we supress noise amplification by the lidar ratio bounds. So, the L-BFGS-B iteration is stopped after a predefined number of iterations of 40.000, after which the average cost function value per bin is required to be smaller than 1. Usually, this holds true after much less iterations. But in

the spirit of the SCA algorithm, the estimate should fit as close as possible to the signal data and only solve the physical contradictions. Hence, a fair comparison to the standard algorithms is achieved only without an implicit regularization. Furthermore, if the MLE can demonstrate its potential even in unfavorable conditions, this will be a strong argument for its usefulness.

    In the following, two methods are used for error quantification of the estimated state vector $\boldsymbol{x}^*$.

A Monte-Carlo approach is applied to classify the uncertainties in simulation results (Xiao et al., 2020). For this, a sufficient number of realizations of the measurement vector $\boldsymbol{y}_{\text{obs}}$ is generated from the simulation scene. The variation of the SCA and




MLE retrieval then yields a reliable measure of the retrieval variability and the standard error. Monte-Carlo approaches yield the best possible results for simulation cases, but are not feasible on observation data.

The errors on observation data are estimated from a sensitivity analysis around the solution, similar to standard error propaga-
tion in spirit. Therefore, the forward model $\boldsymbol{F}(\boldsymbol{x})$ is linearized about the solution point to obtain the matrix equation

$$\boldsymbol{y} - \boldsymbol{F}(\boldsymbol{x}^*) = \mathbf{K}(\boldsymbol{x} - \boldsymbol{x}^*) \tag{16}$$

with Jacobian $\mathbf{K}$. This relation is reversed by the Moore-Pennrose pseudoinverse to obtain a generalized inverse $\mathbf{K}^{-1}$ and hence the sensitivity of the state estimate under changes in the measurement, see Appendix C. With this, the retrieval error covariance matrix $\mathbf{S_{x^*}}$ is calculated and rescaled in accordance with the lidar ratio constraint, see Appendix C.

### 3.3   Aeolus End-to-End Simulation

The algorithms are validated in simulations, which are processed with the end-to-end simulator described in Reitebuch et al. (2018b), which allows realistic simulations of ALADIN data downlinked from Aeolus. Its output data are provided in the same format and temporal and spatial resolution as nominally downlinked from the satellite in order to test the whole processing chain up to the optical properties delivered in the L2A product. The simulation covers the charge transfer and detection on the
accumulation charge coupled device (ACCD) including offsets, non-linearity and noise sources such as dark current noise, read-out noise, Poisson detection noise (shot noise) and the analog-to-digital conversion with 16 bit. Although perfect agreement with the instrument on board cannot be achieved, the simulated noise level approximately resembles nominal operations.

### 4   Results and Discussion

In this section, the two versions of the SCA and the MLE algorithms are tested on synthetic and real Aeolus observation test
cases, and their performances are discussed. The simulated data are produced with the Aeolus end-to-end simulator described in (Reitebuch et al., 2018b), which allows realistic simulations of ALADIN measurements from defined atmospheric scenes as input to the L1B algorithm. Its output data are provided in the same format and temporal and spatial resolution as nominally downlinked from the satellite in order to test the whole processing chain up to the optical properties delivered in the L2A product. The simulation covers the charge transfer and detection on the accumulation charge coupled device (ACCD) including
offsets, non-linearity and noise sources, such as dark current noise, read-out noise, Poisson detection noise (shot noise) and the analog-to-digital conversion with 16 bit. Although perfect agreement with the instrument on board cannot be achieved, the simulated noise level approximately resembles nominal operations. Four simulated and real Aeolus observation test cases have been defined as follows

1. Atmospheric simulation case I: An horizontally homogeneous aerosol profile

2. Atmospheric simulation case II: Case I with an additional cloud

3. Real data case I: A Saharan Air Layer (dust plume) above Cape Verde





4. Real data case II: A ground based comparison with a Polly XT Lidar in Tel-Aviv

In the following sections, the results from each test case are presented.

### 4.1 Atmospheric Simulation Case I

The simulated Aeolus observations from this test case is obtained from the End-to-End simulator and contains horizontally homogeneous aerosol with a constant lidar ratio of 25 sr and the calibration data is known from seperate simulations of Aeolus' calibration modes. A standard atmosphere temperature and pressure distribution is used for the simulation of the molecular backscatter with altitude and the wind speed is zero for simplicity. The simulated atmospheric scene contains aerosol up to 40 km altitude, paticularly above Aeolus' range of about 0-20 km. This aerosol attenuates the useful signals by additional 0.9%,

but no impact is expected on the optical properties since all algorithms allow for a constant attenuation factor.

The retrieval results for the simulation is shown for the SCA, SCA midbin and MLE algorithms in Fig. 2. Please note the logarithmic scales of the colorbar. The medium aerosol load in the input atmosphere with backscatter coefficients on the order of 1 to 10 $\text{Mm}^{-1}$ $\text{sr}^{-1}$ below 2 km is captured well by all algorithms (rows 1 and 2 of Fig. 2). However, clear background noise patterns are visible in the thin aerosol regime with particle backscatter coefficients of order 0.1 $\text{Mm}^{-1}$ $\text{sr}^{-1}$ above 2.5 km for

all retrievals, as expected due to low Mie channel SNR (rows 1 and 2 of Fig. 2). Hence, there are many noise induced negative values in SCA and SCA midbin retrievals. Additionally, SCA and SCA midbin backscatter profiles are shown to be biased in this regime. The MLE mean profile resembles the true backscatter profile, mitigates negative values and consistently shows smaller standard deviation as SCA and SCA midbin.

The coupled retrieval in the MLE unfolds its whole potential in the extinction retrieval (rows 3 and 4): The curtain plots

for SCA and SCA midbin do not only suffer from intense background noise, but also from negative values within the dense aerosol close to the ground, whereas the MLE achieves a much more robust retrieval when compared to the simulation input. The mean profiles unveil high biases in the SCA case: Especially changes in vertical range bin thickness impose a challenge and are followed by extinction overestimation, due to the feedback between zero-flooring of extinction coefficients within the processing and decreased SNR in thin bins. A detailed explanation of the noise influence on the SCA extinction retrieval can

be found in Flament et al. (2021). SCA midbin is less biased, despite a spurious oscillation at about 10 km altitude, but to the price of high noise and lowered resolution. The MLE retrieves only slightly biased extinction coefficients over the whole profile with standard errors up to a magnitude smaller than SCA midbin product. Only the extinction in the bin closest to the ground appears overestimated, likely due to the diminishing influence of lowermost optical depth on the cost function: Here, the generalized averaging kernel $\mathbf{K}^{-1}\mathbf{K}$ deviates from the identity matrix and suggests that this value cannot be well retrieved

(see Appendix C).

The Lidar ratios in row 6 (lowest panel of Fig. 2) are calculated from $\text{mean}(\alpha_p)/\text{mean}(\beta_{||,p})$ and $\text{median}(\alpha_p)/\text{median}(\beta_{||,p})$ of rows 2 and 4 and their respective standard deviation. Otherwise, the first guess of $\gamma_{||,p} = 60$ sr would contaminate the statistics for MLE, e.g., $\text{mean}(\alpha_p/\beta_{||,p})$ would be biased towards the first guess. The lidar ratio results (rows 5 and 6) indicate that the MLE is most robust, as it is the only algorithm for which the lidar ratio statistics converge to the true value of 25 sr everywhere.

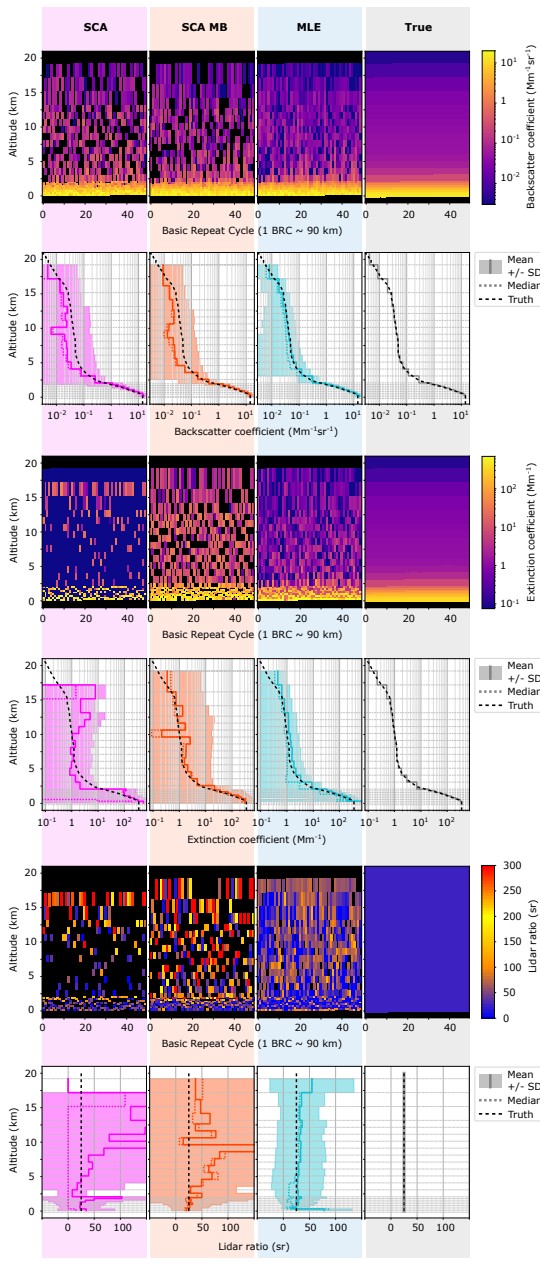

**Figure 2.** Simulation case I: Optical properties retrieval results. Applied algorithms (from left to right) are SCA, SCA midbin, MLE and the true simulation input (rightmost). Curtain plots (row 1,3,5): Dark blue values may exceed the lower colorbar limit and black color indicates negative (or missing) data. Statistics (row 2,4,6): Mean profiles (solid lines) ± standard deviation (shaded) and median (dotted) obtained from 1000 BRCs.





SCA and SCA midbin achieve this only in a narrow range close to the ground. It is also noted that the MLE lidar ratio remains close to 60 sr in scenes with very low signal where the algorithm is not converging. The SCA and SCA midbin produces wither very high (yellow to red values) or zero or negative (black) values in these cases.

## 4.2   Atmospheric Simulation Case II

In simulation case II, the aerosol profile is the same as for case I, but in addition a cloud is placed between 8.5 and 10.5 km
with an optical depth of 0.4, i.e., the return signals from below the cloud are more than halved by a slant two way transmission of $T^2_{\mathrm{cloud}} \approx 0.38$, see Fig. 3. The discontinuity of the cloud (abrupt changes in optical properties) introduces some artifacts into the retrieval results as can be seen in Fig. 3: Most prominently, the vertical extent of the cloud is overestimated by SCA midbin due to the averaging of neighbouring bins onto a coarser resolution (rows 1 to 4). Hence, the cloud thickness appears to be 4 km compared to 3 km in SCA and MLE and 2 km in simulation input parameters. Furthermore, the SCA extinction coefficient
reacts delayed compared to the backscatter coefficient, leading to that the attenuation by the first 500m of the cloud is not captured correctly with consequences for lidar ratio estimation (row 3 and 4 in Fig. 3). The SCA extinction coefficient values below the clouds are biased high due to the abrupt change in signal error. Hence, the same feedback loop as described in the previous simulation case I for changes of range bin thickness is triggered.

The curtain plots of the SCA and SCA midbin optical properties (rows 1,3 and 5) reveal more noise induced negative values
(black) below the cloud and MLE provides the most reliable backscatter estimates below 2 km altitude (row 1 and 2). The statistics below the cloud are heavily impacted by the signal loss: Extinction coefficient estimates below 2 km are found to be biased high in SCA and SCA midbin results and so MLE returns the best fit to the simulation input parameters (row 4). The lidar ratio results (rows 5 and 6) underline that, although the dense aerosol below 2 km is in parts captured by all algorithms, MLE returns the most accurate estimates with highest precision. Hence, the potential of MLE to cope with highly noisy data
compared to the standard approaches is well demonstrated. Again it should be noted that the MLE lidar ratio remains at 60 sr in scenes with very low signal where the algorithm is not converging. The SCA and SCA midbin produces wither very high (yellow to red values) or zero or negative (black) values in these cases.

In summary, the coupled MLE locks extinction and backscatter to appear colocated and so consistently outperforms the standard approaches in terms of both, accuracy and precision. These are graphically summarized for the interested reader in the
Appendix D in terms of bias and relative error, respectively.

## 4.3   Real data case I: Classifying a Saharan Air Layer with Aeolus

In this section, the algorithms are tested and compared using real Aeolus observations from June 30 2020, during the Aeolus satellite overpass close to Cape Verde between 7:30 and 7:40 UTC. The satellite passed above an extended Saharan Air Layer (SAL, Prospero and Carlson (1980)) containing significant amounts of desert dust. The spatial extent of this dust layer is visible
in observations of the UV Aerosol Index reported by the Copernicus Sentinel-5p TROPOMI instrument (ESA, 2018) between 12:00 and 15:30 UTC. Later that day, NASA's Cloud-Aerosol Lidar and Infrared Pathfinder Satellite Observation (CALIPSO) (Winker et al., 2003) made another overpass over Cape Verde between 15:20 and 15:30 UTC which crossed Aeolus ground



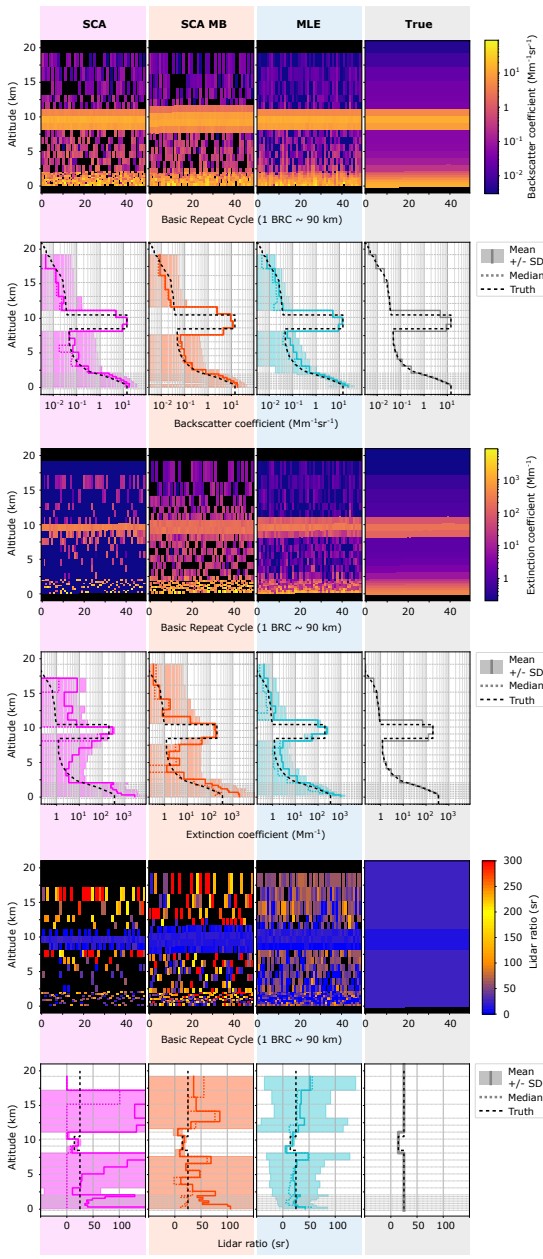

**Figure 3.** Simulation case II: Optical properties retrieval results. Applied algorithms (from left to right) are SCA, SCA midbin, MLE and the true simulation input (rightmost). Curtain plots (row 1,3,5): Dark blue values may exceed the lower colorbar limit and black color indicates negative (missing) data. Statistics (row 2,4,6): Mean profiles (solid lines) ± standard deviation (shaded) and median (dotted) obtained from 1000 BRCs.





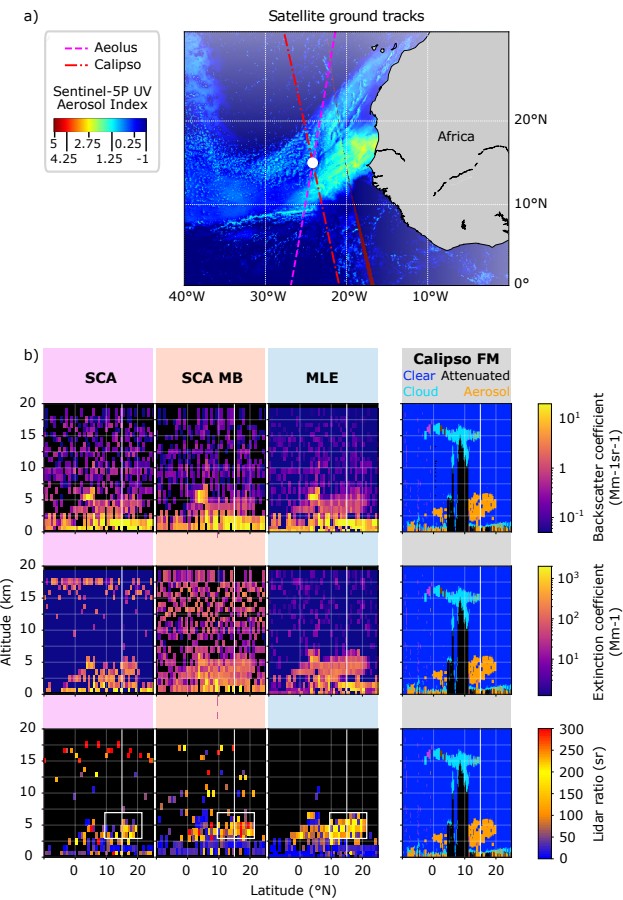

**Figure 4.** Real data case I: a) Map of Africa's east coast overlaid with the ground tracks of the Aeolus satellite between 7:30 to 7:40 UTC (pink, dashed) and the Calipso satellite between 15:20 and 15:30 UTC (red, dash-dotted). Overlaid in color is Sentinel-5p UV aerosol index from 388 nm and 354 nm spectral bands. b) Optical properties processing results for the extended desert dust plume (Saharan Air layer, SAL) from 30th June 2020 for the different algorithms. Columns in panel b (from left to right) are: SCA, SCA midbin and MLE. The rightmost column shows the Calipso Featuremask (FM) for backscatter (first row), extinction (second row) and lidar ratio (last row). Aeolus and Calipso ground tracks cross at the white line. For better interpretability, the lidar ratio is only shown where the corresponding backscatter coefficients exceed 0.25 Mm-1 sr-1 and extinction coefficients exceed 3.75Mm-1.

track, passing over the SAL, see Fig. 4a. CALIPSO carries the polarization sensitive dual wavelength attenuated backscatter lidar, CALIOP. The fairly uniform and large SAL layer is an ideal case to compare the CALIPSO feature mask product (v4.20, data release version 4.10) and the different Aeolus optical properties products. The publicly available L1B data product of baseline 1B10 is used as input to the L2A optical properties prototype processor version 3.12.


The SCA, SCA midbin and MLE processing results for backscatter, extinction and lidar ratio at a horizontal resolution of 1BRC≈87 km along track distance are shown together with CALIPSO Feature Mask (FM) results in Fig. 4b. The afternoon





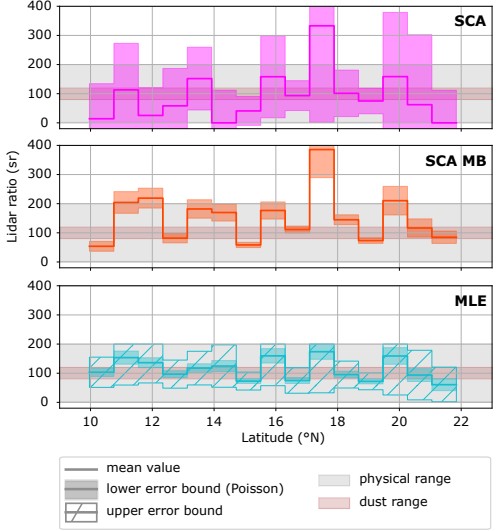

**Figure 5.** Real data case I: Averaged Lidar ratios according to $\langle\alpha_p\rangle/\langle\beta_{||,p}\rangle$ along the Aeolus ground track, calculated from the $4 \times 15$ bins (altitude $\times$ latitude) within the white box in Fig. 4. Applied algorithms (from top to bottom) are SCA, SCA midbin and MLE. The errors from Poisson assumptions are displayed in shaded colors. The upper error bound (hatched) is calculated from the measurement variability, see Appendix B and C. Additionally, the expected range for co-polarized lidar ratio of dust is indicated in shaded brown.

Calipso FM results show a partly lofted aerosol plume that is classified as desert dust in the latitude band 10°N to 20°N, which also compares well with the TROPOMI observations in Fig. 4a. The area towards the equator is partly attenuated by a high ice cloud between 10 km to 16 km altitude, which was possibly on-top of a convective cloud tower. The dust plume likely extended below as more dust is reported by the CALIPSO FM close to the equator. In the planetary boundary layer (PBL) low, broken clouds are visible in the FM. Additionally, patches of clouds are visible, e.g., at about 5 km to 6 km height and 5 °N. The optical properties processing results of the Aeolus signals show similar features, except for the high cloud, which was not present during the Aeolus morning overpass. It should also be noted that convective activity is low in the morning and at its maximum in the early afternoon. Pronounced background noise patterns are present in SCA and SCA midbin backscatter coefficients (noise magnitude about 1 Mm$^{-1}$ sr$^{-1}$) and extinction coefficients (noise magnitude about 30Mm$^{-1}$), very similarly to the simulation cases I and II. Therefore, SCA and SCA midbin show the dust plume only with little contrast and it might be horizontally and vertically disrupted by noise. What is likely clouds is visible by high return signal (yellow spots in backscatter) and low lidar ratios (dark blue color). In these bins with high signal-to-noise ratio, SCA and MLE give very similar results, as expected. SCA midbin suffers from the lower vertical resolution and significantly enhances the cloud and aerosol plume layer thickness, both in the PBL and at about 5 km height, but reconstructs fairly homogeneous extinction coefficients where the plume is located. Much higher contrast to the background is achieved with MLE, so the plume top and plume bottom heights can be inferred along the latitude. The plume detected by Aeolus algorithms agrees well with the CALIPSO FM results despite of Aeolus' coarse resolution and possible representativeness errors due to the difference in time and space between the observations by



the two satellites. Only MLE captures the partly lofted nature of the plume north of the co-location point (white line). Furthermore, MLE achieves more homogeneous results along-track for all properties, which demonstrates robustness and consistency, because no such smoothness constraint has yet contributed to the reconstruction.

Care has been taken in determination of the co-polar lidar ratio of the dust in Fig. 5. All bins that lie within the white rect-
angle in Fig. 4b have been considered for the estimates, due to the fairly homogeneous appearance in MLE and CALIPSO FM. It is not reasonable to directly average the lidar ratio over all bins, as our convention would bias the result: Often, the backscatter-to-extinction ratio (BER) is reported instead, but different conventions produce different results, because $\langle \alpha_p / \beta_{||,p} \rangle \neq \langle \beta_{||,p} / \alpha_p \rangle^{-1}$ with mean $\langle \ldots \rangle$. Hence, we first average backscatter and extinction coefficients over the box and report an unambiguous value for lidar ratio, namely

$$\gamma_{||,p} = \langle \alpha_p \rangle / \langle \beta_{||,p} \rangle \tag{17}$$

that can be transformed into BER. The results read SCA : 78 sr , SCA midbin : 120 sr and MLE : 104 sr which are in line with the expected values of 80sr to 120sr for depolarizing desert dust. This expected range is additionally visualized in Fig. 5, which illustrates the noise induced variability of the Lidar ratio along the plumes horizontal extent: The standard approaches SCA and SCA midbin fail to indicate a coherent feature along track by means of lidar ratio. The estimates range from 0 sr to 320 sr
and 50 sr to 370 sr, respectively, although the results have been averaged over 4 km altitude, i.e., 4 range bins in this case. The MLE achieves the most coherent results along the SAL with estimates between 60 sr and 170 sr and all values lie well within the estimated error ranges. In combination with the curtain plots in Fig. 4, this demonstrates that the presented MLE approach allows for more robust aerosol classification based on the lidar ratio estimates.

### 4.4 Real data case II: Ground-based Validation

On 9th November 2019, Aeolus passed close to a ground based, remote-controlled multiwavelength-polarization-Raman lidar (Polly) in Tel Aviv at 3:50 UTC which is part of PollyNET (Baars et al., 2016; Engelmann et al., 2016). This network consists of several such automated Polly lidars for automated and continuous 24/7 observations of clouds and aerosols around the world. As such, they can also provide vital input for calibration and validation activities of Aeolus' optical properties observations. The presented ground based lidar data in Fig. 6 has been accumulated over the time 2:41 to 3:40 UTC. The distance to the
centers of the two closest Aeolus observations (which are 87 km averages along the satellite orbit) are 68 km and 84 km. A special range bin setting is operational in this area: The altitude range between 2 km and 4 km is divided into 8 range bins of 250 m width to detect lofted dust at highest possible instrument resolution. Thus, the SNR in this layer is smaller than usual, which provides a challenging test case for validation of MLE against the standard approaches. The atmospheric scene was characterized by a temporally stable aerosol layering. For the entire night over Tel Aviv, a PBL with high backscatter
values (3-6 $Mm^{-1}$) up to 1 km was observed while above moderate backscattering ( about 2 $Mm^{-1}$) up to 3.5 km was present. According to the PollyNET target categorization (Baars et al., 2017), both layers were identified as a mix of pollution and dust with particle linear depolarization ratio values of about 10% and lidar ratio values between 40 and 50 sr (at 355 nm). The optical properties from SCA, SCA midbin and MLE processing are overlaid in Fig. 6. The results of backscatter coefficients





appear disrupted by noise for SCA and SCA midbin: Hence, negative or missing values are present in all profiles within the
aerosol layer below 3.5 km but for MLE processing. The MLE is also able to retrieve the vertically coherent Aerosol layer in
good agreement with the ground truth (Raman method at 355nm, Ansmann et al. (1992)). Additionally, MLE results for both
independent Aeolus observations agree well with each other and show much lower variability than the SCA and SCA midbin
results. All aspects considered, the good agreement with the ground case and the consistent retrieval in both neighboring
columns are strong indicators that the MLE successfully supressed noise.

The extinction retrieval is challenged by the lowered SNR in the fine vertical range bin setting, but MLE achieves the most
reasonable results closest to the ground truth and supresses outliers above 3.5 km. After all, the coherent aerosol layer cannot
be well located by using extinction coefficients alone due to too fine range bin settings between 2 km and 4 km that cause high
noise amplitudes. Still, the averaged lidar ratio is calculated from the altitude range 1.5 km to 3.5 km and both observations, as
suggested in equation (17). The results read SCA : $(75 \pm 98)$ sr, SCA midbin : $(73 \pm 31)$ sr and MLE : $(51 \pm 11)$ sr, so MLE
retrieves the most accurate and precise result compared to the ground truth value between 40 and 50 sr. It should be stressed
that these errors represent lower error bounds calculated from the Poisson assumption on measurement noise.

## 5   Conclusions

The optical properties retrieval within the Aeolus Level 2A aerosol optical properties data product has been rephrased into a
MLE problem and was successfully implemented and tested. The evaluation of the new MLE retrieval revealed predominantly
positive impact: It is demonstrated that the precision of the MLE outperforms SCA and SCA midbin algorithms in synthetic
homogeneous scenes and, additionally, reduces biases in the mean statistics. Furthermore, these trends have been confirmed on
two cases of real Aeolus data. All cases consistently indicate that MLE is particularly advantageous for estimation of extinction
coefficients and co-polarized lidar ratios. However, also the precision of co-polarized backscatter coefficients increased overall
as well by application of MLE. This is mainly due to the introduction of lidar ratio constraints that force particle extinction to
appear only where there is particle backscatter along the atmospheric column and vice versa. With this, anti-correlated noise in
the cross-talk corrected signals can be traced back and is effectively supressed, see equations (A7) and (A8) in the Appendix.
The coupled retrieval of all aerosol optical properties (extinction, backscatter and lidar ratio) is in line with the work of Povey
et al. (2014) and distinct from the approaches described in Shcherbakov (2007); Marais et al. (2016); Xiao et al. (2020). Oth-
erwise, the backscatter coefficient retrieval quality could not be enhanced.


A remaining shortcoming of the SCA and MLE algorithms is the signal accumulation on coarse horizontal scales of 87
km although measurements are principally available on 3 km horizontal resolution. If SNR was sufficient, the MLE could
be applied on finer scales as the horizontal resolution alters only the number of atmospheric profiles within the minimization
problem (see equation 15). In order to guarantee sufficient SNR, it would be advantageous to include a feature mask into the
processing chain. With this, the MLE could be performed on distinct features with sufficient signal accumulation. But any
attempt to create a feature mask from the lidar signals itself is flawed, because even a homogeneous patch of atmosphere can





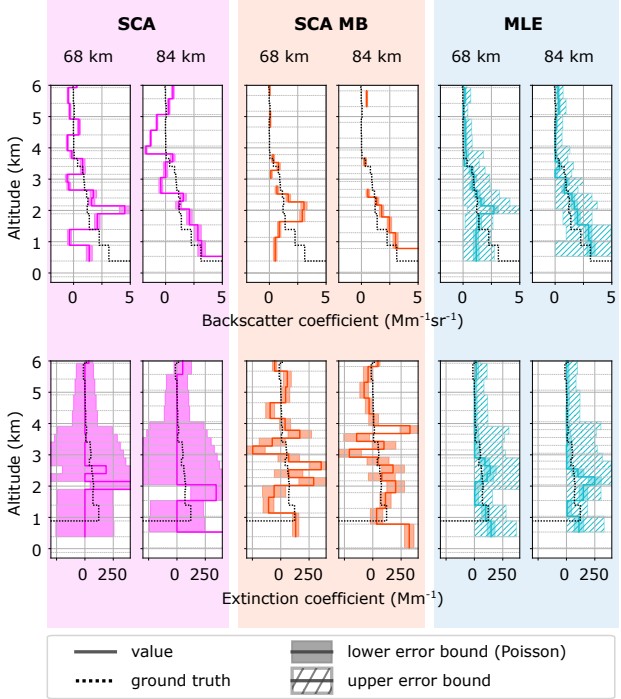

**Figure 6.** Real data case II: optical properties of two co-located Aeolus observations (solid colored lines) compared to the ground truth observations over Tel-Aviv (solid black lines). The ground truth co-polarized backscatter and extinction have been binned to Aeolus resolution. Applied algorithms (from left to right) are SCA, SCA midbin and MLE. Distances between ground station and center of Aeolus observation are 68 km and 84 km. The errors from Poisson assumptions are displayed in shaded colors. The upper error bound (hatched) is calculated from the measurement variability, see Appendix B and C.

appear inhomogeneous (in the signals) when parts of the profiles have been attenuated above. Therefore, in order to estimate aerosol optical properties on finer scales, it is more elegant to apply a suited regularization within the retrieval problem itself, i.e., to add an additonal regularization term in the cost function (see equation 15). Shcherbakov (2007) applied a Thikonov

smoothness constraint in the vertical direction, but stressed the need for a regularization that favours piecewise constant functions. Marais et al. (2016) and Xiao et al. (2020) have therefore made use of the Total Variation regularization in HSRL lidar ratio retrieval along both, horizontal and vertical directions. This is particularly useful to retrieve piecewise constant functions and, hence, the optimal choice for the lidar ratio variable, when the atmosphere is assumed to consist of patches of the same aerosol species. In summary, such regularization is principally able to merge (i) the problem of defining a feature mask from

raw lidar data and (ii) the optical properties retrieval in low SNR conditions. Eventually, we propose to extend the current MLE approach by a Total Variation regularization term in order to enable robust optical properties retrievals with Aeolus on finer scales. This could be attempted in future versions of the MLE algorithm.





*Code and data availability.* The software code of the full L2A processor is not publicly available. The developed procedure/plugin for MLE
processing can be shared upon request to the corresponding author. Data that was used to compile the plots can as well be shared upon
request.

## Appendix A:  Standard Correct Algorithm (SCA)

All details of the SCA are thoroughly described in the algorithm theoretical baseline document of the L2A processor (Flamant
et al., 2017), but this section briefly recaps the main steps and discusses the shortcomings of the SCA. In the following, the
index $i \leq n = 24$ as subscript to the properties above denotes the range bin index. This implies for the signals $s$, $X$ and $Y$
that the property has been integrated over a discrete range $[R_{i-1}, R_i]$, i.e., $s_{\mathrm{ray},i} = \int_{R_{i-1}}^{R_i} s_{\mathrm{ray}}(r)dr$. For all other variables
like backscatter coefficients $\beta$, extinction coeffcients $\alpha$ and range $R$ this subscript denotes the average in range bin $i$, i.e.,
$\beta_{||,p,i} = \frac{1}{\Delta R_i} \int_{R_{i-1}}^{R_i} \beta_{||,p}(r)dr$ with $\Delta R_i = R_i - R_{i-1}$ and equivalently for subscript $m$. As a consequence, particle optical
depth of a bin is denoted $L_{p,i} = \alpha_{p,i} \Delta R_i$ and the particle one way transmission becomes

$$T_{p,i} = \exp\left(-L_{p,sat} - \sum_{j=0}^{i-1} L_{p,j}\right) \tag{A1}$$

with unknown optical depth $L_{p,sat}$ in between telescope and first range bin. The following approximations for the range
corrected signals are made by using the mean bin properties from above (Flamant et al., 2017):

$$X_i \approx \frac{\Delta R_i T_{m,i}^2 \beta_{m,i}}{R_i^2} e^{-L_{m,i}} \left(\frac{1 - e^{-2L_{p,i}}}{-2L_{p,i}}\right) \cdots$$

$$\cdot \exp\left(-2L_{p,sat} - 2\sum_{j=0}^{i-1} L_{p,j}\right) \tag{A2}$$

$$Y_i \approx \frac{T_{m,i}^2}{R_i^2} e^{-L_{m,i}} \left(\frac{1 - e^{-2L_{p,i}}}{-2\gamma_{||,p,i}}\right) \cdots$$

$$\cdot \exp\left(-2L_{p,sat} - 2\sum_{j=0}^{i-1} L_{p,j}\right). \tag{A3}$$

With this, equations 7 and 8 can be rephrased to

$$s_{\mathrm{ray},i} \approx K_{\mathrm{ray}} N_p E_0 \left[C_1 X_i + C_2 Y_i\right] \tag{A4}$$

$$s_{\mathrm{mie},i} \approx K_{\mathrm{mie}} N_p E_0 \left[C_4 X_i + C_3 Y_i\right]. \tag{A5}$$

This way, the signals in Rayleigh and Mie channel are expressed as functions of aerosol optical properties proxies $\gamma_{||,p,i}$ and
$L_{p,i}$ (and $L_{p,sat}$) solely, which can be rephrased into / are equivalent to $\alpha_{p,i}$ and $\beta_{||,p,i}$ (and $L_{p,sat}$). Temperature, pressure
and Doppler shift (HLOS wind) are assumed known a priori by means of ECMWF medium range weather forecast. Hence, the
molecular optical properties, i.e., all variables with subscript $m$ can be inferred with sufficient accuracy as well. The standard
SCA solves equations (A2) to (A5) exactly in a recursive scheme beginning with default initial conditions in the first range bin





to correct for non-zero optical depth above the measurement volume. More precisely, backscatter is calculated directly from

$$\beta_{||,p,i} = Y_i \beta_{m,i} / X_i. \tag{A6}$$

And optical depth / extinction is retrieved independently from the profile of $X_i$. Afterwards, the slant optical depth is transformed into the nadir optical depth by a trigonometric correction factor from which extinction is directly calculated.

But the SCA algorithm suffers from several shortcomings. Any input data is assumed noise-free and an exact solution of the ill-posed extinction retrieval problem is calculated. Among the former strategies to dampen the noise within SCA are averaged extinction estimates over vertically neighbouring range gates, so-called mid-bin properties, and zero flooring, whenever negative extinction values would be obtained in the regular case. These SCA midbin properties are also denoted by SCA MB in this work. Both strategies are ad-hoc but averaging is believed to introduce less bias than flooring, because noise is cut off only in one direction by the latter. Furthermore, the signal noise properties can change abruptly from one range bin to another, due to the varying vertical range bin heights (250 m to 2 km). Hence, if this varying reliability of the signals is not taken into account, biases or oscillations can potentially be triggered whenever range bin heights change. Additionally, extinction is retrieved independently from backscatter, after the linear equation system in (A4) and (A5) has been solved via standard methods. But due to the linear transform, the errors in properties $X_i$ and $Y_i$ are found to be highly anti-correlated. To illustrate this, one can solve equations (A4) and (A5) to obtain from error propagation that

$$\varepsilon_{X,i} = \tilde{C}_3 \varepsilon_{\mathrm{ray},i} - \tilde{C}_2 \varepsilon_{\mathrm{mie},i} \tag{A7}$$

$$\varepsilon_{Y,i} = -\tilde{C}_4 \varepsilon_{\mathrm{ray},i} + \tilde{C}_1 \varepsilon_{\mathrm{mie},i} \tag{A8}$$

with noise terms $\varepsilon$ and rescaled coefficients $\tilde{C}_{1\ldots4} > 0$. Accordingly, the noise from the Mie and Rayleigh channels contribute to the noise in $X_i$ and $Y_i$ with alternating signs so that the negative cross-correlation $\langle \varepsilon_{X,i} \varepsilon_{Y,i} \rangle < 0$ is obtained. Thus, whenever a negative extinction coefficient is found in bin $i$, this indicates a spurious estimate of the backscatter coefficient as well (although it might appear to be well in physical limits) and vice versa. This additional knowledge is not accounted for within the processing and, hence, low SNR in one of the signal channels has the potential to deteriorate both backscatter and extinction estimates. E.g., the contribution from the Mie channel in particle free atmosphere is pure noise, which affects the cross-talk corrected molecular signal $X$ and consequently the extinction retrieval as well.

## Appendix B: Uncertainty Estimates from Measurement Variability

The lidar data on 2.9 km measurement scale (indexed $m$) within an observation is horizontally accumulated onto observation scale of 87 km before the analysis, i.e.,

$$s = \sum_m^{30} s_m. \tag{B1}$$


In case that the scene is truly homogeneous, $s_m$ have the same mean and standard deviation $\sigma_{const} = \sigma_{s_m} \, \forall \, m$. Then, a more robust estimate of the signal uncertainty can be provided from error propagation

$$\sigma_s^2 = \sum_m^{30} \sigma_{s_m}^2 = 30\sigma_{const}^2 \tag{B2}$$

and $\sigma_{const}$ can be estimated from the variance of the 30 signal values on measurements scale, namely

$$\sigma_{const}^2 = \mathrm{var}(s_m) = \frac{1}{30-1} \sum_m^{30} (s_m - \frac{1}{30}\sum_m^{30} s_m)^2. \tag{B3}$$

This estimate will cover all additionally known and unknown noise sources and coincides with the Poisson noise hypothesis in case that the measurement scale signals are truly Poisson distributed. The measurement error covariance $\mathbf{S}_y$ comprises the terms $\sigma_s^2$ for both channels, Mie and Rayleigh, and all range bins on its diagonal.

**Appendix C:  Retrieval Error Estimates**

The errors on observation data are estimated from a sensitivity analysis around the solution, similar to standard error propagation in spirit. Therefore, the forward model $\boldsymbol{F}(\boldsymbol{x})$ is linearized about the solution point to obtain the matrix equation

$$\boldsymbol{y} - \boldsymbol{F}(\boldsymbol{x}^*) = \mathbf{K}(\boldsymbol{x} - \boldsymbol{x}^*) \tag{C1}$$

with Jacobian $\mathbf{K}$. In an optimal estimation method (OEM) framework, this relation is inversed by the gain matrix $\mathbf{G}$, which
contains both the inverse measurement and inverse *a-priori* covariance matrices (Rodgers, 2000). The influence of the latter diminishes to zero in the MLE limit of infinite *a-priori* variances. I.e., no *a-priori* knowledge is imposed, the cost function reduces to the MLE case and $\mathbf{G} = \mathbf{K}^{-1}$. In general, $\mathbf{K}$ will have some zero singular values so that some state vectors cannot be inferred from the measurement. Disregarding these states, this relation can be reversed by the Moore-Pennrose pseudoinverse to obtain a generalized inverse $\mathbf{K}^{-1}$ and hence the sensitivity of the state estimate under changes in the measurement. Thus,
the equivalent to the OEM averaging kernel $\mathbf{A} = \mathbf{GK}$ for the presented MLE reads $\mathbf{A} = \mathbf{K}^{-1}\mathbf{K} \neq \mathbf{I}$ with identity matrix $\mathbf{I}$. This generalized averaging kernel can indicate spurious components of the state vector: Wherever $\mathbf{A}$ deviates clearly from the identity matrix, values will be flagged. But no information on spatial resolution can be inferred from $\mathbf{A}$ as opposed to the OEM. With this, the uncertainty of MLEs retrieved state is phrased in terms of the covariance matrix

$$\mathbf{S}_{x^*} = \mathbf{cov}(\boldsymbol{x} - \boldsymbol{x}^*) = \mathbf{K}^{-1}\mathbf{S}_y\mathbf{K}^{-\mathsf{T}} \tag{C2}$$

assuming that the solution $\boldsymbol{x}^*$ is the true state in the last equation. Hereafter, the bounds can be accounted for as follows: The forward model is rephrased into the $(\alpha_p, \beta_{||,p})$-formulation (see earlier) and the diagonal elements of $\mathbf{S}_{x^*}$ are denoted as variances $\sigma^2$, which form a vector of standard deviations $\boldsymbol{\sigma}$. Now the inequality $\sigma_{\alpha,i} < \gamma_{max}\sigma_{\beta,i}$ is used to constrain the usually overestimated particle extinction error in accordance with the box constraints. Similarly, $\sigma_{\beta,i} < \sigma_{\alpha,i}/\gamma_{min}$ constrains





the particle backscatter error but is usually met by the results already. After this re-scaling of the diagonal elements, the off-
diagonal elements of $\mathbf{S}_{x^*}$ can be re-normalized accordingly under the premise that the corresponding correlation matrix $\mathbf{R}_{x^*}$
remains unchanged. These operations are summarized by

$$(i) \qquad \mathbf{R}_{x^*} = \mathrm{diag}(\boldsymbol{\sigma})^{-1}\mathbf{S}_{x^*}\mathrm{diag}(\boldsymbol{\sigma})^{-\mathsf{T}}$$

$$(ii) \qquad \sigma_{\alpha,i} = \min(\sigma_{\alpha,i}, \gamma_{max}\sigma_{\beta,i}),$$

$$\sigma_{\beta,i} = \min(\sigma_{\beta,i}, \sigma_{\alpha,i}/\gamma_{min})$$

$$(iii) \qquad \mathbf{S}_{x^*,\mathrm{constrained}} = \mathrm{diag}(\boldsymbol{\sigma})\mathbf{R}_{x^*}\mathrm{diag}(\boldsymbol{\sigma})^{\mathsf{T}} \qquad (C3)$$

## Appendix D: Biases and Errors in the Retrieval of the Simulation cases

The following figures provide dedicated information about biases and relative errors in the optical properties retrieval of the
two simulation test cases.

*Author contributions.* Frithjof Ehlers is the main author of this paper. He has developed the MLE implementation and has tested it together
with the SCA algorithms on the test datasets presented here. He has also performed the scientific analysis. The colleagues at Météo France
have developed the Aeolus L2A algorithms and are currently maintaining the L2A processor including important calibration updates and
analyses. As such, Thomas Flament, Alain Dabas, Dimitri Trapon and Adrien Lacour have supported with many fruitful discussions about
Aeolus' calibration procedures and noise characterization. Adrien Lacour provided the end-to-end simulations. Holger Baars has provided
the ground truth lidar data and reviewed this paper. Anne Grete Straume has defined the tasks and workplan for this project and has supervised
Frithjof Ehlers during his German Traineeship at ESA. She has also reviewed this paper.

*Competing interests.* The authors declare to have no competing interests.

*Disclaimer.* The presented work includes preliminary data (not fully calibrated/validated and not yet publicly released) of the Aeolus mission
that is part of the European Space Agency (ESA) Earth Explorer Programme. This includes wind products from before the public data release
in May 2020 and/or aerosol and cloud products, which had not yet been publicly released. The preliminary Aeolus wind products will be
reprocessed during 2020 and 2021, which will include in particular a significant L2B product wind bias reduction and improved L2A
radiometric calibration. Aerosol and cloud products became publicly available in June 2021. The processor development, improvement and
product reprocessing preparation are performed by the Aeolus DISC (Data, Innovation and Science Cluster), which involves DLR, DoRIT,
ECMWF, KNMI, CNRS, S&T, ABB and Serco, in close cooperation with the Aeolus PDGS (Payload Data Ground Segment). The analysis
has been performed in the frame of the Aeolus Data Innovation and Science Cluster (Aeolus DISC).





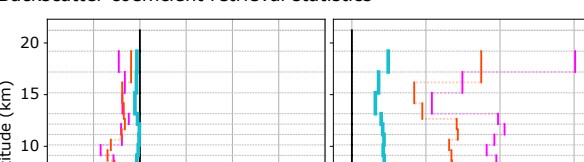

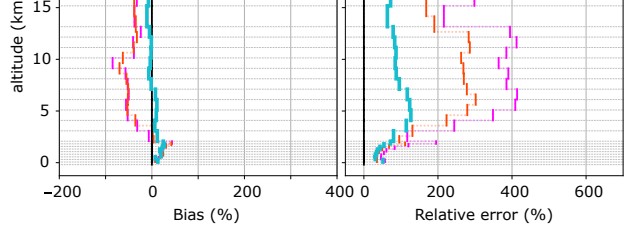

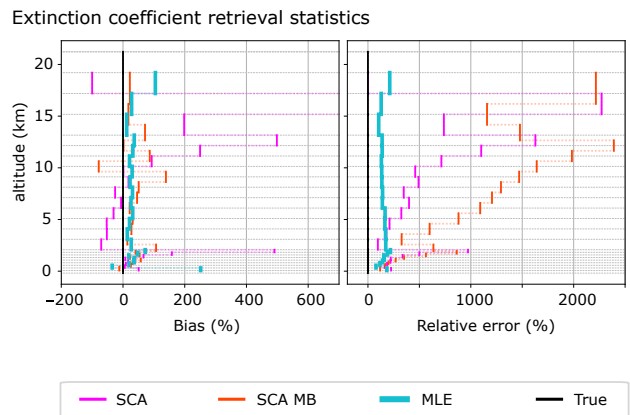

**Figure D1.** Simulation case I: Retrieval statistics for backscatter and extinction coefficients obtained from the mean profiles from Fig. 2. Left: Bias relative to simulation input parameters. Right: Standard error relative to simulation input parameters. Please note the varying x-axis scales.

*Acknowledgements.* The main author has performed this work in the frame of his German Traineeship at ESTEC, enabled and supported by the German Aerospace Center (DLR). His special thanks go to Anne Grete Straume-Lindner and Thorsten Fehr for their dedication and great personal support within extraordinary circumstances. Parts of this research have been supported by the German Federal Ministry for Economic Affairs and Energy (BMWi) under grant no. 50EE1721C. Authors also acknowledge support through ACTRIS-2 under grant agreement no. 654109 from the European Unions Horizon 2020 research and innovation program, and through PoLiCyTa from the German 605 Federal Ministry of Education and Research (BMBF) under grant no. 01LK1603A.



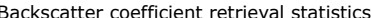

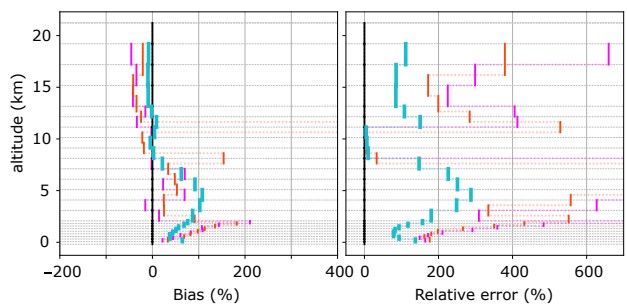

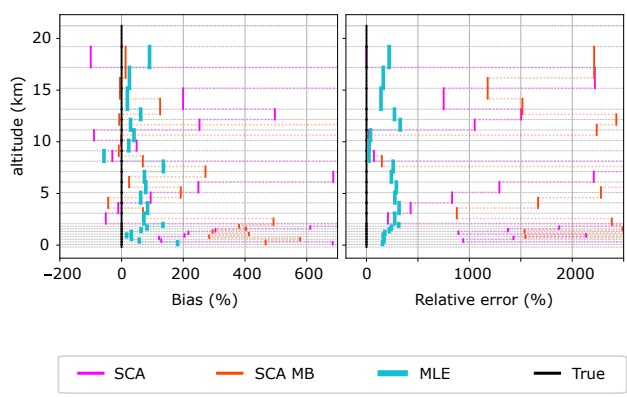

**Figure D2.** Simulation case II: Retrieval statistics for backscatter and extinction coefficients obtained from the mean profiles from Fig. 3. Left: Bias relative to the simulation input parameters. Right: Standard error relative to the simulation input parameters. Please note the varying x-axis scales.

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
