# Peer review of "Optimization of Aeolus' Aerosol Optical Properties by Maximum-Likelihood Estimation"

_Atmospheric Measurement Techniques, 2021_

## Author Response (AR1)

Answer to all Reviewers

First thing to note is that we were very glad about the thorough feedback of all Referees, who evidently made the effort to go through the details presented in our manuscript. This helped a lot to refine the text and we hope that we could make our statements more concise now. Especially the method of calculating the uncertainties in the MLE method was addressed by more than one of the Referees and required some changes that we want to present to all of Referees.

The procedure outlined in the last paragraph of Appendix C was relying on the simple equation alpha = gamma * beta with extinction coefficient alpha, lidar ratio gamma and backscatter coefficient beta. Motivated by comparing the obtained variances via (C2) with the variances one would get from a standard error propagation with constant lidar ratio sigma_a^2 = gamma^2 * sigma_b^2, we argued e.g. that the uncertainty in alpha cannot exceed 200 sr times the uncertainty of beta in a single bin, whatever the precise value of gamma is in reality, due to the upper bound in lidar ratio.

Now, after the feedback and some graphical considerations, the problem with this statement is, that it leaves aside any uncertainty or spread of the lidar ratio estimate itself, as it can lie somewhere in between 2 sr and 200 sr and hence should have non-vanishing variance sigma_g^2, causing another term to arise from the view of standard error propagation, increasing the uncertainty of alpha proportional to beta itself. This is why we noticed that the presented approximation was not correct, although we observed uncertainty values that matched approximately the SCA mid bin output for the SAL case for backscatter and extinction coefficients, which seemed realistic at first glance.

Following this, we evaluated the errors from equation (C2) directly against the uncertainties calculated by the SCA standard error propagation as in Flamant et al., 2020, and found good agreement, when the same Poisson noise variances are used in the $S_y$ in (C2), see attached figure.

[Figure]

That means, that with (C2) we are only able to provide uncertainties for the unconstrained problem (as SCA solves it), but especially the so-obtained extinction uncertainties are overestimated and not representative of the real uncertainties, as one can see in the comparison of SCA and MLE in the simulation cases.

In lack of a representative alternative for uncertainty estimation we decided to delete the lidar ratio and extinction uncertainties of the MLE solutions in Figures 5 and 6, for now, stating that there is no representative error estimate available yet. The challenge of reporting representative uncertainty estimates is now being stressed in the abstract and conclusion and will require future investigation.

**Review 1**

Review of "Optimization of Aeolus Optical Properties Products by Maximum-Likelihood Estimation"

This paper outlines the application of maximum likelihood estimation to the analysis of profiles observed by the high spectral resolution lidar mounted on ESA's Aeolus satellite. To facilitate a bounded optimisation, the lidar equations are cast in terms of layer optical depths and lidar ratios while measurement uncertainties are approximated by the variance of the observations. The method is compared to a more traditional lidar analysis techniques for both real and simulated observations and is shown to produce smoother, more physically consistent fields that are more consistent with other measurements and the truth, respectively. This work, inspired by unexpected limitations in the instrument after launch, hopefully represents the beginning of more routine use of modern retrieval techniques in the analysis of lidar profiles.

I recommend this paper for publication after minor corrections and thank the authors for providing a genuinely enjoyable read on a rainy morning. My only significant comments relate to the discussion of iterations from lines 278 to 292:

- In Eqn. 15 you say $L_p > 0$ but on L281 you say you start iteration at $L_p=0$, which is then out-of-bounds. Presumably you meant to say $L_p >= 0$. Regardless, I am surprised you chose to start at one end of your solution space. I would have started at some climatological mean value to keep the number of iterations down by giving the method the ability to increase or decrease $L_p$ in the first step.

This is correct, we meant to state L_p >= 0. It has been corrected. It is also probably not the best choice to initialise on the boundary of the solution space, but given that the climatological value should be rather small, it does presumably not make a big difference.

- Forty thousand is a preposterous number of iterations! Are you sure L288 shouldn't say 40? Every optimization routine I've worked with tends to converge in 5 to 10 iterations unless the model is extremely non-linear. If you need more than several dozen iterations to get useable solutions, I would guess you've done something wrong – either a gradient is being miscalculated or an inappropriate optimization method is being used.

  When just one profile is considered, as in problem (14), then the number of iterations is on the order of 50 (roughly 0.5 seconds on an office laptop) to achieve convergence (an average cost function value per bin below 1). In practice, however, we are interested in solving problem (15) for all profiles simultaneously. In this case about 5000 iterations are required (roughly 45 seconds on an office laptop) to generate good optical properties products for a whole orbit (roughly 450 profiles). This information has been added to the manuscript.

  So essentially, solving (15) for the whole orbit saves us time. Though, on second thought there must be a better converging formulation, but we have had no reference on this. We made an attempt to reformulate our model in terms of log(L_p), but the convergence speed did not increase noticeably. It is very likely that a more advantageous scaling exists since the L-BFGS-B algorithm is not scale invariant. Unfortunately we haven't found it.

- I don't agree with you at L291 that artificially extending the number of iterations produces a fairer comparison to the SCA. This sounds like an attempt to avoid discussing the method of identifying convergence because those are contentious. I honestly don't care how you do it, but I do think the paper should explain how you intend to produce data outside of the context of this validation and, ideally, give some idea of the magnitude of difference. I suspect it makes very little difference, which is why you ran a set number of iterations, but that should be stated.

  You are right that for an operational use, the presented approach is not handy. So we decided to add at the end of the paragraph: "For operational use we plan to tune the number of iterations in an ad-hoc-fashion, based on if the average cost function value per bin has fallen below a value of 1."

In contrast to OEM and regularized MLE, the only contribution to the MLE cost function is how well the raw signals y are represented by F(x*). Once they are within the estimated limits, we can essentially stop iterating, because we cannot expect to gain any additional information content from the measurements. So this will determine convergence.

- Something similar comes up again at L357 when you say the first guess "contaminates" your mean. A first guess is not a prior; it should not affect your final solution. Exceptions are where the method fails (and so you throw away the solution) or there are multiple minima (which would require a more advanced minimisation routine). Could you explain what you mean here? Is it just that you aren't doing any quality control and so failed retrievals are present in the output? If so, you should do quality control! You don't need a value from every single pixel.

So the problem is the following: Whenever the retrieved aerosol optical depth L_p is very low, the influence of the lidar ratio estimate on the cost function becomes increasingly insignificant, until L_p becomes zero and no lidar ratio can be provided at all (these are not included in the statistics). Wherever L_p is very low, the lidar ratio has a high error estimate and some may remain close to the first guess, which then seems to act as an 'implicit a priori' (see https://doi-org.tudelft.idm.oclc.org/10.1029/JD095iD05p05587, end of section 7). What our paragraph wants to say is that, if lidar ratios were averaged without weights, then the first guess would contaminate the statistics. In the shown simulation case, this would only affect the statistics when only bins with retrieved backscatter coefficients below $5*10^{(-2)}$ (Mm sr)$^{(-1)}$ are considered.

So eventually we propose to replace the sentence "Otherwise, the first guess of $\gamma_{\||,p}=60$ sr would contaminate the statistics for MLE, e.g., $\text{mean}(\alpha_p/\beta_{\||,p})$ would be biased towards the first guess." with "[...], in order to disregard the influence of bins with nearly vanishing aerosol optical depth, for which no reliable lidar ratio can be retrieved."

Additionally, lines 360-361 and lines 380-381 are dropped, because they are more confusing than helpful.

The following comments are mostly matters of personal curiosity rather than issues that need to be addressed:

- The text in figures should aim to be approximately the same size as the text in its caption. If it is possible, all the figures in this paper would benefit from being regenerated with a smaller page size, such that the font is larger relative to the image.

The font size of figures 2,3,4 and D1, D2 has been adjusted.

- L68) I would say backscatter is 'measured' rather than 'known' with higher precision.

  corrected

- L77) Is there a reason you preferred to constrain L to be positive rather than retrieve its logarithm? I could see this being useful in your proposed future work to produce consistent regularization in variables that span several orders of magnitude. Further, aerosol optical depth (which is what you actually retrieve) is known to be log-normally distributed, such that the log of L is a more natural basis in which to define a regularization.

  There is actually no elaborate reason for the choice of L instead of log(L), but it might contribute to proper scaling of the variables, indeed. We will consider this aspect in future work.

- L103) I feel like this paper demonstrates that such a retrieval is possible rather than it being something you need to assert in advance.

  The possibilities were discussed pre-launch and the SCA has been implemented already before. The paper's scope is improving the precision of the retrieval.

- L204) I've not encountered this use of a backslash before. Did you mean to say "a ratio"?

  No, we meant "or". It has now been dropped for convenience.

- L230) While true, lidar signals are often far from this limit.

  Considering also the comment of Reviewer 2, we changed the sentence to

  "[...] because the discrete Poisson noise distribution can already be well fitted by a smooth Gaussian with identical mean and variance for very low (photon) counts and the aforementioned additional noise sources and their corrections, e.g., subtraction of measured solar background, will naturally smear out the discrete nature of the Poisson noise."

- L261) I think that the two uncertainty ranges shown in Fig. 5 come from the two sources of uncertainty you mention here – a simple Poisson assumption and the variance of the downlinked profiles – but it would be useful for that to be stated clearly somewhere. This sentence currently implies the Poisson

approximation is not used in the remainder of the paper, but you go on to mention it several times.

Thanks for underlining this source of incovenience. The Poisson approximations in Fig 5 and 6 have only been used for the uncertainty estimation and not in the minimization. Following the feedback of Reviewer 2 and 3, we noticed that our implementation of the box-constraints in the error calculations was flawed. In the lack of an alternative we removed what we initially claimed to be an upper bound in Figures 5 and 6.

We also edited the last paragraph of 3.2 to explain that the uncertainties of SCA and SCA MB are calculated as in section 6 of the Algorithm Theoretical Baseline Document for the L2A and changed the name of this uncertainty to "Poisson error estimate" in Figures 5 and 6.

- The first paragraph of Section 4 unnecessarily repeats the preceding paragraph.

  The paragraph 3.3 has now been deleted.

- I would mention the existence of Appendix D at the end of L343 as that's when I asked "what's the RMS deviation"?

  According to the feedback of Reviewer 3 we promoted a modification of figure D1 (with zoomed in boxes for better visibility below 2 km altitudes) to the main text and added a more detailed discussion of the biases and relative errors in the regime of most interest below 2 km altitude.

- In the caption for Fig. 4, don't you mean the west coast? Also, the description of 4(b) implies that three versions of the feature mask are shown. I think you meant to say that the rows show backscatter, extinction and lidar ratio, with identical features masks shown in each frame of the rightmost column.

  We corrected for both errors.

- L418) I'd argue that the image shows the advantage of forcing L>0 rather than robustness.

  This sentence lacked a concise message indeed, so we changed it to say "[...] which demonstrates the advantage of the box-constraints."

- L493) Is there any intention to release this data? Any possibility of funding to process the full record or become an operational product?

  The MLE is being considered for operational implementation.

- Appendix A largely repeats Section 3.1.

- L516) ECMWF data would typically be interpolated rather than averaged as the values represent behaviour at a point rather than a grid cell average. What do you mean by "mean" here?
  We write "by means of" in the sence of "thorugh" or "with the help of" and do not refer to the statistical mean.

I found a number of grammatical corrections:

- L2) the Atmospheric Laser
  How embarrassing.

- L2) is an Ultra Violet
  corrected

- L5) Being an HSRL
  corrected

- L10) demonstrate a predominantly
  done

- L11) information by the SCA
  corrected

- L15) due to effective noise
  corrected

- L27) respectively, and the Cloud-Aerosol
  corrected

- L32) addressing high uncertainties in climate change modelling due to the indirect
  We changed this sentence to "[...] for the largest single cause of uncertainty in antropogenic radiative forcing has been reported to be from the indirect effect of aerosols on clouds (Illingworth ...)"

- L46) [commas around 'and similarly Raman lidar observations']
  corrected

- L48) increase the SNR
  corrected

- L68) extinction
  corrected

- L90) Either "revolves around", "orbits", or just "Aeolus is in a Sun-synchronous polar orbit"
  corrected to "revolves around"

- L90) seven day repeat
  corrected

- L92) Earth with an off-nadir
  corrected

- L93) Traditionally, "Earth" is the planet and "earth" is dirt so I don't see why you use both versions here.
  Although amusing in its current form, we corrected the non-capital "earth"s.

- L93) There should be a space between numbers and their units.
  corrected

- L125) [remove the comma]
  corrected

- L135) scales dependent on
  corrected

- Fig 1) accumulation to observation
  corrected

- L164) [comma after 'following']
  corrected

- L172) into a molecular
  corrected

- L178) [commas around 'in principle']
  corrected

- L181) can be simplified by introducing the range
  corrected

- L209) Appendix A.
  corrected

- L281) consist of an aerosol free
  corrected

- L296) y_obs are generated
  corrected

- L302 and 563) Moore-Penrose
  corrected

- L312) the real instrument cannot
  This sentence has been altered otherwise.

- L334) particularly
  corrected

- L334) by an additional
  corrected

- L336) simulation are shown
  corrected

- L350) but at the
  corrected

- L381) produces very
  corrected to either

- L409) cloud
  corrected

- L428) the lidar ratio along the plume's horizontal
  corrected

- L439) The ground based
  corrected

- L450) aerosol
  corrected

- L457) be located by using extinction coefficients alone due to the fine range
  corrected

- L484) additional
  corrected

- L487) [delete comma]
  corrected

- L514) channels
  corrected

- Eq A7) [something has gone wrong with the spacing]
  corrected

- L541) e.g.
  corrected

- L559) is inverted by
  corrected

- L561) i.e.
  corrected

I largely appreciated the relatively conversational tone of this manuscript compared to the average paper, making relatively dry material easier to read. But there are a few places where I would have made a different choice of words. As these are a matter of style, I leave it to the authors to decide if they agree with me or not.

- I don't think "Products" is necessary in the title of the paper. Then again, I would have called it "Optimization of Aerosol Optical Properties from Aeolus Profiles by Maximum-Likelihood Estimation" because I'm concerned that someone might think the "Optical Properties" are possessed by Aeolus rather than by particles in the atmosphere.
  This is a very good point, also considering that the Journal's scope can be on both measurement systems and the data analysis. We agree to change the title.

- Remove "(partly)" from L8. A problem is either ill posed or it isn't; there is no partly.

  We removed ill-posed as what we mean is "sensitive to or ambiguous under noise influence".

- L52) A white paper under preparation in the UK is hoping to call these "representation errors" rather than worry about 'representivity' vs 'representativeness'. Obviously there's no obligation to agree with us but I think it's a cleaner phrase.

  Fair enough, we follow this advice since the term "representativeness" seems to bulky anyhow.

- L122) "corrected for to obtain" doesn't look strictly wrong but sounds very strange to a native near; I wouldn't say the 'for'.

  corrected

- Fig 1) While this image is technically "Exemplary" in that you are using it as an example, English has corrupted the word to typically mean "Outstanding". I'd say "Illustrative" or "Example of" instead.

  Thanks! We did indeed not mean to exaggeratedly underline the presented data.

- Eq 2-3) I know you're using T for transmission but it hurts slightly to see temperature denoted t. I would have used \mathcal{T} for transmission.

  I agree but this way the difference is clearer and comes without unnecessary flourish. If we had used temperature frequently, we would have changed it for sure.

- L179) Maybe "extensive" rather than "excessive" as the latter has negative connotations.

  Thanks! We dropped the word completely.

- L312) Perhaps "noise level resembles nominal" as 'resembles' already implies an inexact comparison.

  agreed

- L370) "...reacts after the backscatter coefficient, resulting in the attenuation of the first 500 m of the cloud being captured incorrectly with consequences..." I struggled to understand this sentence and have made my best guess at what you were trying to say.

  We changed it to "the SCA extinction coefficient reacts delayed compared to the backscatter coefficient, leading to that the attenuation by the first

500m of the cloud is not captured correctly".

- L373) "Hence, the same feedback loop is triggered, as described in case I for changes of range bin thickness." I find it difficult to understand long sentences where the verb is at the end, so I moved the verb to the start.
  agreed

- L383) "MLE causes extinction" as 'locks' is a rather strong word for what happens.
  Agreed, we changed it to "forces"

- Fig 3) This caption could just say "As Figure 2 for Case II".
  corrected

- Fig 4) Is it necessary to use very similar colours for the two lines? If you moved away from a rainbow colour bar, you would have more options so I don't have to strain to see if the line has little dots in it or not.
  Our excuses, but this data has been plotted through online services. I completely agree that rainbow color bars are no optimal choice.

- L463) Perhaps "refactored" rather than "rephrased"? The latter usually refers to words rather than actions.
  We chose "reformulated" instead.

- L479) Considering you go on to critique the use of a feature mask, you might want to say "it is possible to" rather than "it would be advantageous to".
  Thanks for making this statement more consistent.

- Eq C2) I have never seen the notation $K^{-T}$ to indicate an inverse transpose. Probably because it's ambiguous if I transpose the inverse or inverse the transpose. $(K^T S_Y^{-1} K)^{-1}$ is a more traditional way to write this equation.
  Indeed true, thanks. We changed it to $(K^{-1})^T$ instead.

- I was taught to put a hyphen between words that together form a single adjective. I provide a non-exhaustive list of places where you could do that: L33 & 439) ground-based; L90) seven-days; L95) Rayleigh-Brillouin; L101) double-edge; L123) time-gated; L137 & 140) so-called; 281) aerosol-free; L374) noise-induced; L393) polarization-sensitive, dual-wavelength; L541) particle-free.
  Thanks a lot, we corrected for all occurrences of these terms as most of them were mentioned several times throughout the paper.

**Review 2**

The authors propose an alternative extinction coefficient retrieval methodology for the ESA's Aeolus ALADIN instrument that performs better compared to the current Standard Correct Algorithm (SCA). The authors formulated the retrieval problem as a maximum likelihood estimation (MLE) problem in which the ALADIN HSRL forward models are incorporated. The benefit of the MLE methodology is that it allows for constraining the physical values of the parameters that are being estimated (e.g. backscatter coefficient and lidar ratio).

The reviewer is disappointed that the authors did not expand on the work of Xiao et al. 2020 and Marais et al. 2016 who both introduced MLE methods that employ the total variation (TV) image smoothness constraint to improve the inference of the extinction coefficient for Poisson noisy observations. The publication would have been much better if the authors at least applied the MLE TV method on spaceborne lidar data. There are readily available software implementations of the MLE TV method, one of which is called SPIRAL and it is available at http://drz.ac/code; the authors will be required to make modification to the SPIRAL code, but that should be within the expertise of the authors.

The reviewer, however, is recommending a minor revision since the authors are improving the state of the art specific to the Aeolus ALADIN instrument measurements.

Major comments:

Lines 81 to 83: In the references that are cited, the work were specific to developing inversion methods for ground based lidar instrument. The authors are proposing a spaceborne inversion HSRL method. This distinction matters because for ground based lidar instruments inaccuracies in the geometric overlap function calibration parameter can introduce biases in the extinction coefficient. Hence, the backscatter coefficient is inferred as a separate first step to isolate any geometric overlap function induced biases to the extinction coefficient. With spaceborne HSRL measurements and depending on the instrument, there is not necessarily a geometric overlap function which are confirmed by equations 2 and 3 in this paper. Thus, a spaceborne HSRL allows for the simultaneous inference of the backscatter and extinction coefficients that are not biased to due to the geometric overlap function.

For the sake of understanding how this paper's method relates to the references that are cited, the reviewer suggests that the authors mention that the geometric overlap function calibration parameter is a limiting factor for ground-based HSRL instrument when trying to simultaneously infer both the backscatter and extinction coefficients.

We acknowledge the reviewers point of view and suggest to add the sentence: "It is important to note that such a simultaneous retrieval with ground based lidars would require the additional geometric overlap function calibration parameter. Hence, it is often preferred to retrieve backscatter coefficients independent of extinction coefficients to mitigate biases on the former due to calibration errors."

Line 198: It is unclear where in (Flamant et al 2017) equations 9 and 10 are defined. Since equations 9 and 10 are esoteric, it would be helpful for the reader to know where in (Flamant et al 2017) equations 9 and 10 are defined since (Flamant et al 2017) has 124 pages.

We are aware that these equations are rather unappealing, but felt the need to state them once in the exact form they are used. We now refer to the equivalent equations by number in the cited document: "[...] see equations 6.35-6.36 in Flamant et al. (2017) and definitions above."

Lines 199-200: Equations 9 and 10 do not make sense. For example, consider equation 9. Since $L_{p,i} > 0$, we have that $-2L_{p,i} < 0$. Hence, $X_{i} < 0$, since the rest of the terms in equation 9 are positive. However, this a contradiction, since $X(r)$ in equation 5 is strictly positive.

Indeed the minus in the denominator was incorrect so it was removed in in (9) and (10) and (A2) and (A3). For your readability: The function $(1-\exp(-2x))/(2x)$ is almost equal to $\exp(-x)$ for small x.

Line 224: Should J be $J_{obs}$? Should $S_y$ not be the inverse of the measurement error covariance matrix? Refer to equations 2.21 and 5.3 of (Rodgers, 2000).

Thanks for spotting these mistakes, we corrected for them throughout the whole manuscript and added the suggested references after line 224.

Line 287: The reviewer does not agree with the statement "MLE estimates usually suffer from overfitting and noise amplification.", because it is the forward model along with the MLE that determines whether there is overfitting or "noise amplification". E.g., the averaging operator (or just averaging) can be derived via MLE and averaging operator with a sufficient number of samples does not overfit. In other words, whether a MLE overfits or does "noise amplification" depends on the MLE's parameterization with the forward model.

This is a good point and was certainly expressed with too little care in the manuscript. We now changed the wording to "MLE estimates may suffer from overfitting and noise amplification in lidar retrieval problems, so an implicit regularization is often achieved by optimal choice of the number of iterations [...]", while keeping the existing references, in which more information can be found.

Line 230 to 232: Could the authors provide a reference to support the claim that the averaged signal's noise can be approximated via a Gaussian distribution? For example, are there sufficient photon counts during the nighttime at 30km altitude that are accumulated and normalized in order for the average signal's noise (at 30km) to be accurately approximated by a Gaussian distribution?

The reference to the central limit theorem was misleading. What we meant to say is that besides its discrete nature and asymmetry, the Poisson distribution can be decently well approximated by a Gaussian with same mean and variance. So instead we propose to change the old wording
"[...] because the Poisson noise distribution becomes indistinguishable from a Gaussian given sufficient signal accumulation (central limit theorem)."
into
"[...] because the discrete Poisson noise distribution can already be decently well approximated by a smooth Gaussian with identical mean and variance for very low (photon) counts and the aforementioned additional noise sources and their corrections, e.g., subtraction of measured solar background, will naturally smear out the discrete nature of the Poisson noise."

Lines 259 to 261: Poisson noise is not additive and therefore it is confusing to read \hat{sigma}_{s} = \sqrt{s + \epsilon_{s}}. For Poisson noisy observations the noise standard deviation is \sqrt{s + b_s + b_d} where b_s is the solar background radiation and b_d is the dark counts.

The mathematical formulation was meant to support the understanding, but can indeed be misleading here as it was inspired by the perspective of Gaussian noise. We propose to remove this part and include "because a single draw does in general not equal the true mean and the true variance" as a reason for this bias from using the signal samples themselves.

Lines 291 to 292: The statement "the estimate should fit as close as possible to the signal data and only solve the physical contradictions" is contradictory compared to the previous sentences in this paragraph. If the estimate should fit as close as possible to the observations (signal data), then why not let the L-BFGS-B converge to a solution and why bother introduce an early stopping criteria of 40 000 iterations? In other words, in this sentence the authors are suggesting that the estimator should "overfit".

There is no contradiction if read the following way: Since the number of 40.000 iterations is really high (see other reviewer's remarks), we try to maximise convergence by brute force and hence aim to eradicate any implicit regularization additional to the constraints.

Lines 431 to 433: Please add a reference that shows the range of lidar ratio values for desert dust at wavelength 355nm.

This is certainly helpful for the flow of reading. We added "(Wandinger, 2015)".

Lines 457 to 460: Are the authors reporting co-polarized lidar ratios or BERs? If BERs are reported, how were the co-polarized lidar ratios converted to BERs?

No BER is calculated here, but we refer explicitly to the co-polarized lidar ratio now.

Sections 4.1 and 4.2: The authors should validate the uncertainty quantification as described between lines 300 and 305, since it is unclear how accurate the proposed uncertainty quantification is for observational data.

see statement of the author's

Section 4 & conclusion: The authors do not thoroughly explain why SCA methods produce biased results compared to the MLE method, and it will be insightful to know why SCA methods produce biased results. Could it be that the X and Y (equations 7 and 8) are modeled by SCA (equations 9 and 10) could introduces biases in the estimates of the backscatter and extinction coefficients relative to the MLE method, since the MLE method employs equations 7 and 8?

The forward model for SCA and MLE is equivalent. The MLE also relies on eq. 9 and 10, see line 212. So this is not the reason for biases.
The biases occur in our opinion mostly due to the non-linear operations in equations (9) and (10): Take as an example the ratio operation and arbitrary random variables x and y such that mean(x)/mean(y) = 1. Now, depending on the noise amplitudes on x and y, mean(x/y) can be biased high to any value, solely because of the varying noise amplitude. Considering the non-linear character of equations (9) and (10), it becomes explainable that biases in SCA and SCA MB can be (partly) mitigated with MLE by suppressing noise.

Regarding the high bias in the lowermost 2 km we added:
"For the origin of this bias we can think of two causes: Firstly, the violation of the hypothesis of uniformly filled bins due to the strong gradient in aerosol concentration and, secondly, the non-linear way the backscatter coefficient is calculated from $\beta_{||,p,i}=Y_i\beta_{m,i}/X_i$, because here $mean(\beta_{||,p,i})$ will become biased high increasingly with increasing uncertainty of $X_i$."

Comment about the methods: It will also be useful know what is the performance of a method that directly algebraically solve for the lidar ratio and backscatter coefficient via equations 7 and 8. In other words, what is the performance of the MLE method without constraints? The reason why this comparison will be useful, is to gain an understanding of the low performance of the SCA methods. Specifically, is the loss of performance due to the formulation in equations 9 and 10, or because of the lack of physical constraints?

As stated above, both SCA and MLE rely on the same equations (9) and (10). So the SCA is the direct, algebraical solver for the MLE problem, as suggested. Hence, if the MLE had no constraints and was initialised with the same condition of vanishing optical depth in the first bin (as SCA is) then outputs of SCA (without zero flooring of extinction, see figure 10 in https://earth.esa.int/eogateway/documents/20142/37627/Aeolus-Data-Innovation-Science-Cluster-DISC-Level-2A-user-guide.pdf for an example) and MLE would be virtually identical. Indeed we see very similar noise patterns to this figure 10 when applying no constraints in MLE.

We added the following to the text after equation (14):

"It is important to mention that it is only the a-priori knowledge in form of the box-constraints that makes the MLE solution distinct from the algebraic SCA solution (without zero flooring, see section 6.2.2.1 in Flamant et al. 2020), because this algebraic solution corresponds to the exact signal values in y and therefore to $J_{obs}=0$, which is the global minimum to the unconstrained counterpart of problem (14)."

Minor comments:

Line 2: In the first paragraph of the manuscript introduction all the letters of Aladin is capitalized. To be consistent, Aladin should be all in capital letters.

checked

Line 5: "Being and HSRL" should be "Being an HSRL".

checked

Line 6: Backscatter coefficients and lidar ratios are normally reported without polarization.

Aeolus can only directly measure / report these values.

Line 6: It will be helpful to the reader to know that ALADIN does not make cross-polariation measurements (lines 107 & 108) and therefore ALADIN is not able to make direct lidar ratio measurements.

We agree that this might be confusing for the lidar community, hence we added the remark "(the cross-polarized return signal is not measured)".

Line 8: Do the authors mean that the inversion problem is statistically ill-posed? If the inversion problem is ill-posed, then it would not be possible to infer the extinction coefficients from ALADIN measurements without using lidar ratios.

This statement was based on the work of Pornsawad et al., 2008: "But From a mathematical point of view Eq. (1) is a Volterra integral equation of the first kind, which is ill-posed." To be honest, we find this wording misleading as well and thus decided to drop "ill-posed" throughout our text.

Line 33: It is unclear what the authors mean by "because high uncertainties in climate change modelling regard the indirect effect of aerosols on clouds and anthropogenic radiative forcing (Illingworth et al., 2015).". Are the authors saying there are high uncertainties of indirect effects of aerosols on clouds in climate change modelling?

That sentence was a bit bulky. The actual statement in Illingworth et al. reads: "The largest single cause of uncertainty in anthropogenic radiative forcing is from the indirect effect of aerosols on clouds." So we reformulated our text to say "[...]; the largest single cause of uncertainty in antropogenic radiative forcing has been reported to be from the indirect effect of aerosols on clouds (Illingworth ...)."

Line 41-42: Lower SNRs of what? This part of the sentence is vague.

This refers to the actual lidar signals. So the SNR of the signal. In order not to write the "signal's signal-to-noise ratio" we added "in the receive channels" for clarity.

Line 47: Do the authors mean that the inversion problem is statistically ill-posed? If the inversion problem is ill-posed, then it would not be possible to infer the extinction coefficients from ALADIN measurements without using lidar ratios.

See answer to comment on line 8.

Line 69: Fine resolution of what? Image resolution?

We mean the effective resolution in the retrieval, which is not clear. Likewise, any explanation would inflate the text here, so we kept only the statement about the higher precision.

Line 75: "reformulated as a " would read better than "rephrased into a".

done

Line 133: Do the authors mean detector random errors instead of wind random errors?

No, the primary concerns for Aeolus were the wind random errors.

Line 135: Is it unclear what is meant by Basic Repeat Cycle (BRC) or Observation. Are these terminologies that are used in the field of wind lidar? If so, would it be helpful to add a reference? Does BRC mean that every 30 consecutive vertical profiles are accumulated?

These are Aeolus mission terminologies, which are to find in the Aeolus documentation referenced in this section. But maybe it was misleading that we used "integrated"? Instead, we mean "accumulated", which should clarify.

Line 148: "is used as input to the optical" should be "is used as input by the optical".

done

Line 162: The reviewer presume that the measurement geometry is the so called geometric overlap function. It is unclear why the authors define O(r) which it is not used in equations 2 and 3?

We wanted to give a general formulation, but it is true we don't use it. We added "For Aeolus the range overlap function $O(r)$ equals 1."

Line 163: The range of T(r) is defined but not for \beta(r). To be consistent it will be helpful to also define the range for \beta(r) e.g. \beta(r) > 0.

done

Line 168: If the ALADIN instrument is measuring the co-polarized backscattered energy, shouldn't the molecular backscatter coefficient also be labeled as co-polarized?

To my knowledge, it is assumed here that the molecules are a non-depolarizing target, making both the same.

Line 180: Using the word excessive sounds negative; it would be best not to use this word.

done

Line 190: "When" should be "where".

done

Line 203: What is 'this' referring to? Is 'this' referring to L_{p,sat}?

We now explicitly referenced the equations we intend to refer to.

Line 205: Did the authors mean to say "or" instead of "/"?

Yes, we corrected this.

Line 221: It is unclear what the "It" refers tp. Does the "It" refer to the second term in equation 12?

Yes, also corrected

Line 222: Should J be J_{obs}?

Here it is equivalent, but we clarified this in the text.

Line 225: In regards to S_y, refer to comment of line 224.

corrected

Line 226: Should "Lidar" be "lidar"?

corrected for all occurrences

Line 234: In regards to S_y, refer to comment of line 224.

corrected all occurrences

Lines 236 to 245: The paragraph is superfluous since 1) the authors are not employing a constraint or penalty term and 2) the paragraph does not add value to the current text. If the authors at a later stage employ a constraint or penalty term and report results in a next publication, the next publication can include this paragraph. Furthermore, the conclusion does discuss employing a regularization term.

The reviewer suggests replacing this paragraph with one line saying that equation 14 is applied on each averaged vertical profile of measurements.

We acknowledge the reviewer's point of view, but are bound to report our method the way it is implemented. Since the actual implementation hands the problem (15) to the L-BFGS-B algorithm and not (14) in a for loop over single profiles, we would prefer to keep the paragraph.

Line 252: "would show off diagonal" should be "would have off diagonal".

corrected

Line 256: The last sentence on this line is unclear. To what does "following" refer to?

We changed it to "As in SCA, this overlap is not considered in this work."

Lines 263 to 264: Are the 2.9km and 87km numbers referring to different horizontal resolutions?

Yes, exactly, see Fig. 1 for clarification (horizontal length values added).

Line 268: The citation style of Wandinger at at. 2015 is not consitent with the citation style of Illingworth et al. 2015.

corrected

Line 288: It is unclear why new sentence starts with "But".

Pardon, that is a remnant of my German. We coupled the sentences to transport our message.

Lines 306 to 313: Large portions of sub-section 3.3 are repeated in the first paragraph of section 4. Therefore this sub-section can be removed.

Many thanks for spotting this mistake. We removed sub-section 3.3.

Line 362: It is unclear what the authors mean by "wither".

Line 382: It is unclear what the authors mean by "wither".

We meant to say "either"; corrected

Figures 2 and 3: For the curtain plots; what is the horizontal axis? Profile number? Seconds?

This is the profile number, but in Aeolus' mission-specific vocabulary. See Instrument section: "[...] a total of 30 measurements are accumulated to one so-called Basic Repeat Cycle (BRC) or Observation (Aeolus mission terminology), equivalent to approximately 87 km along track distance on ground."

Line 427: Can the authors elobrate on how the cross-polarized lidar ratio is transformed into BER (the actual lidar ratio)?

The cross-polarized BER is meant. This is added to the text now.

Line 434: More robust compared to what?

As compared to the reference algorithms. "than SCA and SCA MB" has been added.

Line 473 to 475: See the comments of lines 81 to 83.

We added: "The choice of dependent and independent retrieval of backscatter and extinction coefficients is a trade-off between improved precision and potential biases. A coupled retrieval may improve the precision of the retrieved backscatter coefficients, but it relies on a potentially erroneous calibration as input (geometric overlap function and cross-talk correction)."

**Review 3**

Review of "Optimization of Aeolus Optical Properties Products by Maximum-Likelihood Estimation" by Ehlers et al.

The article describes a new algorithm for retrieving aerosol optical properties from Aeolus that improves various shortcomings of the existing Aeolus algorithms. Specifically, the new algorithm operates on the direct measurements and their uncertainties (rather than cross-talk corrected pseudo-signals ignoring uncertainty), simultaneously fits both extinction and backscatter (rather than sequentially), and uses explicit constraints to force solutions to have non-negative optical depth and lidar ratio within expected limits (rather than implementing these constraints as filters after the retrieval). The analysis includes the calculation of error bounds that appear to better reflect actual uncertainties than those of the standard algorithm. Comparisons between the new and existing algorithms are presented for simulated scenes and real data scenes where correlative lidar measurements are available.

I have an overall favorable impression of the manuscript and look forward to seeing it published. The improvements over the existing algorithms are useful and justify the publication of this study. An appropriate level of detail is presented and the logical flow of the manuscript is good and helpful for understanding the algorithm and results. However, there are some important aspects which need clarification and revision, particularly those relating to the uncertainty calculations. Since I see these as critically important, I am recommending major revisions, despite the overall high quality of the manuscript.
* * *
Major concerns

P3 L77. A positivity constraint on quantities where zero is a valid and common expected value will produce a bias. The authors mention this in the context of the SCA algorithm, but this is also a concern in the new algorithm. It might be better not to include the positivity constraint on backscatter and extinction in the new algorithm. Have the authors tried their MLE method without the positivity constraints?

Yes, we tried this. The MLE method without any constraints shows no advantage over the SCA, since the observed signal values can be perfectly fitted to. Without the

zero-flooring, the SCA produces oscillatory extinction results as in Figure 10 in https://earth.esa.int/eogateway/documents/20142/37627/Aeolus-Data-Innovation-Science-Cluster-DISC-Level-2A-user-guide.pdf. Comparably bad solutions are obtained using MLE without constraints.

We added after equation  (14):

"It is important to mention that it is only the a-priori knowledge in form of the box-constraints that makes the MLE solution distinct from the algebraic SCA solution (without zero flooring, see section 6.2.2.1 in Flamant et al. 2020), because this algebraic solution corresponds to the exact signal values in y and therefore to $J_{obs}=0$, which is the global minimum to the unconstrained counterpart of problem (14)."

p10 Box constraints. I think its possible that adding constraints has a smaller impact on improving the results than the other improvements: using raw measurements and uncertaintites in a coupled retrieval.

The forward model equations used by SCA and by MLE are virtually identical and differ only in terms of some algebraic simplifications/variations. So it is indeed the a-priori knowledge in form of the constraints that causes the improvements.

Additionally, the constraints have potential negative consequencs: (1) that they could lead to bias (similar to what happens in SCA) and (2) that they make propagation of the uncertainty very difficult.  Have the authors attempted the MLE retrieval without adding the box constraints?  It would be useful to compare results with and without using the box constraints. If they are just as good, eliminating the box constraints would eliminate the two issues mentioned above. If they are not as good, the comparison would give a clearer view of the impact of the different improvements.

This is a good point and has not been made entirely clear in the manuscript either, but as mentioned above: Solving the MLE without box-constraints is essentially a more complicated way to converge to the algebraic SCA solution (without zero-flooring), which is not favourable.

P19, L469 "This is mainly due". Similar to the previous comment.  This statement is made without any analysis to suggest how the different algorithm features dominate the improvements. The authors should assess the impact of the box constraints separately, to support this statement that it is the dominant cause of improved results.

For the above mentioned reasons, the SCA essentially is this benchmark. So, we do not intend to include an analysis of the "MLE without constraints", as it would just illustrate the oscillatory behaviour outlined in section 6.2.2.1 in Flamant et al. 2017.

P6 Eqs 2 and 3. P21, Eqs A4 and A5.  Very little is said about the cross-talk constants C1, C2, C3 and C4.  The top of page 7  seems to imply the constants are known, but in fact they are very challenging to determine and have significant uncertainty associated with them. More information is needed about how these are determined and what are typical values for Aeolus. This is also important for the discussion of the correlated errors and for understanding the magnitude of that problem.

We agree, the typical magnitudes for C1...C4 should be added. For the calculation of the calibration coefficients please see the L2A ATBD (Flamant et al. 2020, updated 8th Juli) section 6.5 and the stated reference document RD25 "Generation of AUX_CAL Detailed Processing Model Input/Output data definition" available at https://earth.esa.int/eogateway/documents/20142/37627/Aeolus-AUX-CAL-IODD-DPM.pdf .

To add this essential information in the text, we included just above section 3.1:

"The instrument is calibrated with measurements from dedicated instrument calibration modes (Reitebuch et al., 2018a) and the cross-talk coefficients $C_{1...4}$ are determined according to Flamant et al.,2020 and the procedure in Dabas, 2017. At (p,t,f) = (1000 hPa, 300 K, 0 MHz) $C_1$ and $C_4$ equal 1 by definition. The other coefficients then typically take values about $C_2 \approx 0.5$ and $C_3 \approx 1.25$. For the rest of this work, we assume the calibration as known and do not include the contribution of calibration errors in the results. The calibration cannot be perfect in reality, but is likewise input to all algorithms, which guarantees a fair comparison of retrieval precision in Sections 4.3 and 4.4."

We also include in the abstract now:

"The increased precision of MLE with respect to SCA is demonstrated by increased horizontal homogeneity and better agreement with the ground truth on real data cases, though proper uncertainty estimation of MLE results is challenged by the constraints and the accuracy of MLE and SCA retrievals can depend on calibration errors, which have not been considered."

p10-11 L294-304, two approaches to error quantification. **While in general I like the idea of analytical propagation of errors using sensitivity analysis, the propagation is seriously challenged by the difficulty of representing the impact of the constraint in the propagation equation.** I'm not very convinced by this method of rescaling the variance and assuming the correlation is unchanged (Appendix C, near P23 L575). **Can it be explained more clearly why the correlation matrix should remain unchanged?** Also, using the lidar ratio constraint to (potentially) reduce both the extinction and backscatter error bounds separately gives an impression of circularity. The authors should add rigor and validation to make this method more convincing.

see statement of the author's

P25 Figure D1. Although the Monte Carlo methodology is discussed, it seems that the results of the Monte Carlo approach are only shown in Appendix D without any analysis or discussion. These should be promoted to main text and properly analyzed.

The uncertainties in the Figures 2 and 3 are from Monte-Carlo assessment, which is in the main text already. We now refer to it explicitly with having changed line 295 to say "A Monte-Carlo approach as in Xiao et al. 2020 is applied to classify the uncertainties in simulation results in sections 4.1 and 4.2."

Both methods are available for the simulation cases, so I'd like to see a comparison of the analytical (Appendix C) method against the Monte Carlo results for the simulation cases, which might bolster my confidence in the analytic method.

see statement of the author's

It would also be useful to compare the propagated uncertainties with and without the adjustment for the constraints, to see how big this impact is.

Typically, the extinction uncertainty estimate is reduced because the uncertainty of the extinction is usually much higher than 200sr times the uncertainty of the backscatter.

The authors say a Monte Carlo approach is not feasible for the measurement cases, and I think this refers to the fact that they cannot vary the true measurement error. However, it is certainly possible to simulate measurement noise from the measurement error covariance matrix, $S_y$. Using these simulated measurement errors in a Monte Carlo approach would give an accurate view of how $S_y$ is propagated through the retrieval, more immediately convincing than the method in Appendix C. So, going further, I'd like to see a Monte Carlo propagation of $S_y$, which could be used to validate or replace the current propagation.

We cannot follow this point well. What we have is the Forward model $y = F(x)$ going from the state space x to measurement space y. Now, what you suggest is to propagate y errors through $y = F^{-1}(x)$, which contains the implicit function $F^{-1}$. Finding this implicit function locally is the problem we solve time-consumingly via optimization already. So, if we understand correctly, you suggest to estimate the error via multiple runs of the minimization procedure with artificially added noise from $S_y$ to the already noisy signals S? This artificially increases our noise amplitude and makes it necessary to run the minimizer (at least 20 times) more often, which we find to be unfeasible effort for the operational product.

P17, Figure 5 (also figure 6). I can't understand which error bounds are shown in Figure 5 and Figure 6. The descriptions "lower error bound" and "upper error

bound" are confusing because they don't match any description in the methodolgy, and they are possibly inaccurate as well. Another terminology should be seletected, preferably one that reflects the descriptions in the methodology section. The two sets don't seem to be the "two approaches" introduced on P10-11. Are they instead two formulations of the analytic propagation of errors using different characterizations of the measurement error S_y? Is the "upper error bound" the propagated uncertainty from Appendix C?  (If so, calling it an upper error bound is particularly problematic since the Cramer-Rao inequality characterizes a lower bound, not an upper bound). And how is the "lower error bound (Poisson)" calculated?  I don't see that in the manuscript.

see statement of the author's

If indeed the one labeled "upper error bound" is the one produced in this analysis, and the one labeled "lower error bound" is the standard one for SCA and midbin-SCA, then another emergent theme of this analysis is that the uncertainty of the standard algorithms is inaccurate as well. This should be highlighted in the manuscript as another primary impact of the new algorithm.

see statement of the author's

Minor comments

P1 L8 "algebraic inversion scheme to a (partly) ill-posed problem and therefore sensitive to measurement noise".  The sentence should probably be reworked. "(partly) ill-posed" is not really informative, and even a well-posed inversion is sensitive to measurement noise.

This statement was based on the work of Pornsawad et al., 2008: "But From a mathematical point of view Eq. (1) is a Volterra integral equation of the first kind, which is ill-posed." To be honest, we find this wording misleading as well and thus decided to drop "ill-posed" throughout our text.

P2 L53-54 I think this point about there being no resolution between these regimes is particularly well articulated.

You mean, too well articulated?

P3, L78-79. **I think the sentence should be split up into multiple sentences that clearly list the factors that led to poor results in the SCA (there are at least two, I think: the correlated noise and the incorrect removal of negative extinctions)**

and the factors that lead to improvements (coupled retrieval, using the measurement error covariance matrix in the retrieval, box constraints), and to explain their causal relationships. In particular, I believe it is incorrect to suggest that the box constraints are responsible for eliminating the correlated noise issue, as this sentence seems to say. Instead, I think the bias is avoided by using the measurement channels and their error covariance matrix in a coupled retrieval.

We keep this sentence as it is but added the respective reference to the zero-flooring and averaging operations in SCA within Appendix A. Without the box-constraints, the MLE would not 'know' which fraction of any signal value likely corresponds to noise, therefore, in (A7) and (A8) the MLE would be 'blind' and not be able to improve anything.

P4, L130. How is the grid spacing chosen? What are the typical values?

The range bin sizes are typically coarser at high altitudes and finer close to the ground but change several times per orbit and area of interest. For better readability we changed this sentence to "Individual, vertical range bin sizes can be independently varied between 250 m and 2000 m in steps of 250 m, while remaining limited to a total number of ACCD rows and hence vertical range bins of 25 per column."

p7 Eqs 9 and 10. The minus sign in the denominator does not appear in Flament et al 2017. The rest of the equation looks similar but with a few more algebraic steps. I think it is otherwise correct, but I would like the authors to double-check to be sure.

Thanks for spotting this error; the minus sign has been corrected and the equations have been double checked.

p7 Eqs. 9 and 10. It would be helpful to have a sentence or two summarizing the derivation of the quoted equations; for example, an explanation that the equation includes an expression that explictly integrates the transmission over an extended vertical bin.

For better reading flow we changed the line above these equations to say: "The following approximations for the range corrected signals (5) and (6) are made by using the mean bin properties from above, see equations (6.35)-(6.36) and definitions above in Flamant et al., 2020:" We hope that clarifies? We are sorry but we cannot repeat all derivation steps from Flamant et al. as this would not add value to the manuscript.

p8 near the the end of section 3.1, it would be good to have a sentence or two discussing the anti-correlated noise that's mentioned in the Appendix. This seems like an important point that is referred to both in the introduction and later, so it

should be described in the main text with enough detail so a reader knows what it is about.

It is not too important and should remain in the Appendix to compress the main text of the manuscript. The main insight after the algebra should be that the backscatter and extinction both depend on both Sray and Smie at the same time.

P9 L232. I'd like to know more about how the box constraint is implemented and specifically whether it may negatively impact results. Could it potentially bias the results (by perhaps producing a disproportionate number of solutions near the boundaries) or affect their precision of the solution (via a variable transform that changes the gradients near the solution and near the boundaries)?

We are no experts in this either, but these details of the L-BFGS-B algorithm can be found in the work of Nocedal et al. on http://users.iems.northwestern.edu/~nocedal/lbfgsb.html, which is underlying the scipy.optimization package.

p9 242. "future developments". This is really more of a response to another reviewer, but I just wanted to say that I support the authors' decision to perform single profile retrievals without including horizontal scene smoothing. It makes sense to explore the simpler solution first. Furthermore, not requiring scene continuity for the algorithm means that scene continuity can be used as a check on algorithm performance.

We thank you for your remark and fully agree with this position, as otherwise no conclusion could be drawn from the comparisons in Figures 5 and 6.

p9 L250 As I mentioned, I believe that using the measurements and their appropriate measurement error covariance matrix is largely responsible for avoiding the bias due to the correlated errors in the cross-talk corrected signals. Do the authors agree? If so, this paragraph might be a good place to highlight that.

On the contrary, we will need to highlight that the a-priori information in the forms of the constraints is responsible for the noise suppression capabilities. This we hope to have accounted for with the additional text added according to comment on P3 L77.

p10 L272-275. Is there anything in the current algorithm that addresses the bias from partially filled bins? If not, is the current algorithm compatible with the strategy implemented by Flament et al. 2017?

No, we do not follow the strategy of the ICA, that is why SCA is used as a benchmark. While the approach reads promising, the ICA has essentially too many degrees of freedom to yield useful results and is not currently maintained by ESA due to its very poor performance. In the data product, the only columns that seem reasonable are basically almost unchanged copies of the SCA results.

Consequently, no effort has been made to tailor this implementation to the ICA. Already with the SCA, one could criticize that the effective resolution of extinction coefficients is expected to be coarser than the range bin size itself.

p10 L290-292. There appear to be three rather weak arguments here to justify not using an a priori covariance matrix such as called for by OE. **The weakest part of the argument, in my opinion, is saying that the algorithm has no prior information or regularization.** In fact, the constraint is prior information and does provide regularization. If the authors believe the influence of the constraint is significant (which I think they do, because they attribute the improved results largely to implementing the constraint), then even suggesting that this is a weak constraint would not be justified. Therefore, the authors should acknowledge that the constraint plays the role of prior information in the retreival, but they chose not to cast this constraint in the terms required by optimal estimation, because that would require a different (probabalistic) form for the prior that isn't compatible with the desired form of the constraint. **In my opinion, that is a reasonable reason to use constrained MLE rather than OE.** (However, the choice leads to difficulties in working out the correct way to calculate the impact of the constraint on the uncertainties, which I discussed in the "Major" section).

Do you rather mean lines 231-232? We fully agree with your understanding of the matter. What we meant was that we do not have an explicit term in the cost function. We highlight the importance of the box-constraints but also added after (14): "[...] with box-constraints on lidar ratio, which is prior information that cannot be exactly represented by OEM."

p10 L295. Some clarification would be helpful here about the simulated measurement noise. Is it simulated as gaussian with the variance determined from Appendix B, or is it simulated with various separate error components? Which components are included?

The information on which error components on the signal noise the simulation results include can be found in the beginning of section 4. These noise components are to the best of our knowledge modelled realistically, not just added up variances.

p12 L347-349. While I certainly agree that filtering negative results would cause a high bias that is worse in bins with low SNR, I can't clearly follow the explanation in this part of the text and I don't see a particularly clear indicator that they are triggered by the shift from coarser high-altitude bins to finer low-altitude bins as implied by the text. Can the explanation be clarified?

According to Reviewer 2 we promoted a modified version of FIg. D1 into the main text, which we now refer to for clarity of the statement. You can see the locally highest extinction coefficient biases at 2 and 13 km altitudes in Fig. D1 exactly where

the range bin settings change. An exception is the zeroth bin of course, since it is always normalized to zero extinction in SCA.

p12 L354. The indicator from the averaging kernal is a good idea. Is there a quality flag in the data product related to this? It would be nice if this indicator were included in the plots to show which bins are not trustworthy. This would be particularly useful in Figure 4, for instance, where I am curious to know if the bins below the lofted aerosol are reliably retrieved. This curiosity is fueled by the fact that CALIPSO's 532 nm data frequently misses the aerosol at the bottom of attached layers due to attenuation, making them look lofted. The HSRL capability should act to prevent this problem given sufficient SNR, but on the other hand, attenuation at the shorter wavelength of 355 nm would be worse. So, to be sure, it would be good to see some indication from the retrieval algorithm that the lack of aerosol below the apparent plume bottom is reliable.

The values that cannot be retrieved are padded with -1 in the data product. Usually, this regards only the lowermost extinction bin. Regarding the bins below the lofted layer: We do of course see increasing error estimates with increasing distance from the satellite but we fear we cannot say if the layer does only appear to be lofted in both Aeolus and CALIPSO data or if it is lofted for real.

p12 L354. "The extinction bin closest to the ground cannot be well retrieved." Is this true in general, or specific to this case?

True in general, due to the set of the used equations. We added the word 'generally' to the text for clarity.

p12 L357 "Otherwise". I'm confused by this sentence, and not sure if I'm confused by content or just the wording. Does "otherwise" indicate the strategy of taking the mean lidar ratio? I agree that taking the ratio of the mean extinction and backscatter is a better strategy than the mean lidar ratio and will give potentially different results, since lidar ratio for small extinction and backscatter is not as reliable as when the SNR is higher. But why is the mean lidar ratio contaminated by the first guess? And how does taking the ratio of mean extinction and backscatter avoid the influence of the first guess?

This overlaps with a point that Reviewer 1 made as well. So the problem is the following: Whenever the retrieved aerosol optical depth $L_p$ is very low, the influence of the lidar ratio estimate on the cost function becomes increasingly insignificant, until $L_p$ becomes zero and no lidar ratio can be provided at all (these are not included in the statistics). Wherever $L_p$ is very low, the lidar ratio has a high error estimate and some may remain close to the first guess, which then seems to act as an 'implicit a priori' (see https://doi-org.tudelft.idm.oclc.org/10.1029/JD095iD05p05587, end of section 7). What our paragraph wants to say is that, if lidar ratios were averaged without

weights, then the first guess would contaminate the statistics. In the shown simulation case, this would only affect the statistics when only bins with retrieved backscatter coefficients below 5*10^(-2) (Mm sr)^(-1) are considered.

So eventually we propose to replace the sentence "Otherwise, the first guess of $\gamma_{||,p}=60$ sr would contaminate the statistics for MLE, e.g., $\text{mean}(\alpha_p/\beta_{||,p})$ would be biased towards the first guess." with "[...], in order to disregard the influence of bins with nearly vanishing aerosol optical depth, for which no reliable lidar ratio can be retrieved."

Additionally, lines 360-361 and lines 380-381 are dropped, because they are more confusing than helpful.

Figure 2. What is the explanation for the low bias in the median lidar ratio from the MLE for the lowest bins?

The pdf of MLE backscatter is not symmetric due to the positivity constraint. Hence, the median is not equal to the mean.

Figure 2-4 and D1: The line plots are so small that I can't see important information. The data are particularly important below about 2.5 km where there is significant aerosol, but this is a very small portion of the plot and not readable due to its size and the closely spaced grid lines. Specifically, the lowest bin is called out in the text, but it is so small I cannot see the data or error bar for that lowest bin in the line plots. Please include inset boxes to show the line plot data in the lowest 2.5 km, or a second set of line plot figures that show only the lowest 2.5 km, or in some other way improve the visualization of the lowest 2.5 km.

This is a really good point for consideration. So we now adjusted the font size in the mentioned figures to match better with the text and also provide and discuss an overhaul of figure D1 in section 4.1, in which inset boxes make this information on biases and errors accessible.

p14 L362. The reduced uncertainty from the new retrieval compared to the SCA should be discussed and quantified. I think the new retrieval probably produces usefully smaller uncertainty, but the fact that the figures are so tight makes it nearly impossible to see the region where there is significant aerosol below ~2.5 km, and I can't even tell if the error bars are smaller than 100%. A discussion of uncertainty results is just as important as the mean tendency of the profile, because, for example, a profile that "looks right" but is indistiguishable from 0 everywhere due to its uncertainty would be rather useless.

According to the remark above we considered your feedback and added a more detailed discussion in 3.1 by including:

"In order to better illustrate this, the bias

$(\text{mean}(x^{*})-x_\text{true})/x_\text{true}$ and relative error $\text{std}(x^{*})/x_\text{true}$ for all retrievals with respect to the true profile are presented in Fig. D1. Here, the maximum backscatter coefficient bias in the aerosol layer below 2 km is reduced from about 43\% with SCA and SCA MB to 27\% with MLE. This bias seems to be triggered by the refined range bin setting below 2 km. For the origin of this bias we can think of two causes: Firstly, the violation of the hypothesis of uniformly filled bins due to the strong gradient in aerosol concentration and, secondly, the non-linear way the backscatter coefficient is calculated from $\beta_{||,p,i}=Y_i\beta_{m,i}/X_i$, because here $mean(\beta_{||,p,i})$ will become biased high increasingly with increasing uncertainty of $X_i$. The relative error in backscatter coefficients is consistently lower for MLE compared to SCA; In the most interesting area below 2 km the relative error in backscatter coefficients reduces to 50\%-30\% with MLE compared to 120\%-50\% for SCA, while MLE performs only slightly better than SCA MB."

and

"The MLE retrieves the least biased extinction coefficients over the whole profile with standard errors up to a magnitude smaller than SCA midbin product, see Fig. D1. Retrieved extinction coefficients are all biased high in the area below 2 km, with maximum bias of 500\% for SCA, 110\% for SCA MB and 70\% for MLE. Between 1.5 and 0.5 km altitude, the biases are comparable in magnitude with about 30\%. Concerning the relative extinction errors, an improvement by a factor of about 1.5 to 2 in comparison to SCA and SCA MB is achieved by MLE in the lowermost 2 km. Though the relative error is on the order of 100\% or greater in all cases."

P14, L366. Is this number a typo?  exp(-2*0.4) = 0.45 not 0.38.

This has to do with Aeolus viewing angle of about 35 degree off nadir. This means exp(-2*0.4/cos(35deg)) = 0.38 from the instruments point of view.

P14, L369-379. It seems that it's not just delayed (i.e. slow decay below the cloud) but the cloud is smoothed into the regions both below and above.  In other words, it looks like the effective resolution of the extinction is much less than the backscatter, which make sense since it takes at least two measurements to calculate a derivative. Is there any attempt to calculate backscatter on the same coarser resolution as extinction to produce the lidar ratio?

The phrasing of delay regards the curtain plot in row 3 of Fig. 3 in which you can see that the onset of the cloud is positioned at the ground truth for MLE but is delayed by a bin in SCA. We now properly indicate this in the text. The slow decay below the cloud must be a result of the zero-flooring and the noise in SCA, as we cannot see it

in SCA MB and MLE. MLE relies on the same forward model and should therefore have identical effective resolution, but due to the limitations of MLE as compared to OEM, we have not determined any effective resolution.

P15, Figure 3. It appears that the bias in the mean backscatter below the cloud is actually worse in the MLE than in the SCA and SCA-midbin results. This should be discussed.

By promoting figure D2 into the main text, we also added some more detailed discussion and included this observation.

P19, L449, what causes missing values in SCA midbin?

These values are flagged out in the operational processor since they are negative. This is the case only in this figure, but has been suppressed in the generation of the end-to-end simulation statistics. For future reprocessing campaigns it is planned not to flag negative values anymore.

P19, L458, Lidar ratios aren't shown in the figure for the Real Data Case II, as in other cases. Better to show them, if possible.

The individual lidar ratio estimates on single bins are basically so noisy in the SCA cases, that this would be of no value for the reader (with zeros and negative values also due to this backscatter-extinction delay problem), therefore we provide only the calculation. We referred to this noise problem in the text already.

P19, L459-460, Is this comparison between copolarized lidar ratio from AEOLUS to total (co- and cross-polarized) lidar ratio from Polly? This is not the best option. Since Polly is also sensitive to polarization, wouldn't it be possible to calculate copolarized lidar ratio from Polly for a more direct comparison? In any case, it should be clearly stated what's being compared.

The Polly results provided by Holger do mimic the Aeolus range bin settings and the co-polarized lidar ratio that Aeolus sees. We replaced all mentionings of lidar ratio in the text with co-polarized lidar ratio.

P19, L461. It seems strange that the quoted uncertainties are from the Poisson method that the authors are hoping to replace. It would be better to quote uncertainties from the method that the authors think are more represenatative.

see statement of the author's

P 20, Figure 6. Is there an uncertainty bound available for the "ground truth" (which is also a retrieval)?

We are sorry, but this uncertainty has not been provided.

P20, Eqs A2 and A3 are difficult to mentally derive from the previous step. It would be helpful if more intermediate steps were added to make it clearer how the equation is derived. The appendix is a good place to do this.

This line has been edited the same way as in comment on p7 Eqs. 9 and 10.

P22, L538. This point about anticorrelated error is interesting and informative. However, some parts of the discussion are confusing. For instance, Eqs A7 and A8 show that an error spike in one measured channel gets distributed in an anticorrelated way into the corrected signals. But how does it follow that there is correlation (or anticorrelation) in the errors in backscatter and extinction? It seems logical that if the errors in the two corrected channels are anticorrelated, the ratio would tend to be biased low, so the backscatter would tend to be biased low. However, the error in the extinction would not be correlated with it, because extinction derives from gradients in just one of these corrected channels.

What we meant to express here is rather that:

 "This means, e.g., that one noisy value in the Mie (or Rayleigh) channel disturbs both backscatter and extinction coefficients. That also implies that if an unphysical, negative value is obtained in one bin for backscatter (or extinction) in the final result, then the value for extinction (or backscatter) is definitely disturbed as well, whether it lies within physical bounds or not."

This apparent additional information, which lies below the surface so to say, is not used after the SCA has been applied once.

That is an update to the statement:

"Thus, whenever a negative extinction coefficient is found in bin $i$, this indicates a spurious estimate of the backscatter coefficient as well (although it might appear to be well in physical limits) and vice versa."

P23, L557, I suggest replacing "in spirit" with something more informative, such as "except with an adaptation to account for the impact of the constraint"

We decided to drop this phrase.

P23, L561. It's not true that no a priori knowledge is imposed.  The box constraints are a priori knowledge.

Thanks, we changed "a-priori knowledge" to "explicit a-priori term" to be more concise.

P25 Figure D1. "Bias" and "standard error" should be defined in equations.

This has been promoted into section 3.1. See answer to p14 L362.

Grammar and word usage

I found that, although the overall flow and logic are very quite good, in some spots the word choice made the writing difficult to understand. I've marked several below. My list is probably not complete, however, so a round of editing for English usage (not just spelling and grammar) would be helpful.

Title. The title would be more informative if it contained the word "aerosol".  For example: Optimization of Aeolus Aerosol Optical Properties Products by Maximum-Likelihood Estimation

Agreed!

P1 L1.  Typically "embarks" is not used this way, and used only for the much narrower circumstance of a person getting on or off a vessel or starting a journey. It could be replaced with "includes"

We decided for "carries".

P1 L5.  Being an HSRL

Corrected.

P1 L9. Consider replacing "rephrase" with "rework" or "revise"

We decided for "reformulate"

P1 L12. Delete "equally".  I think most writers would use "also" rather than "equally" here, but it really isn't necessary at all.

Has been dropped.

P1 L14.  Consider moving the phrase "to consolidate and illustrate the improvements" to the beginning of the sentence. I find it easier to follow when phrases that modify the verb are near the verb.

Modified.

P2 L32. "Because of high uncertainties" and "regarding the indirect effect" (add "of" and "ing")

This sentence has been modified to be more concise.

p2 L34. Delete "aspect of"

Done.

p2 L34. Consider inserting a paragraph break after "coverage"

P2 L53.  Insert "For Aeolus" at the start of the sentence that begins "There is no suitable resolution"

Agreed.

p3 L64. I suggest deleting the parentheses around "particle"

Agreed.

p3 L69-70. I'm not sure I understand what is meant by "well located".  Does this mean retrieved at finer resolution?  (Perhaps not, because if so, it still doesn't quite make sense.  I understand that extinction and backscatter are retrieved simultaneously on the same grid, but the effective resolution of extinction and lidar

ratio will always be less than the finest possible backscatter resolution because it takes at least two measurements to determine a derivative.)

We modified the sentence to "Thus, the particle extinction may occur only where there is backscatter and [...]"

p4 L94. Delete "means of"

Agreed.

p4 L96. emitted

Done.

P4, L130. Consider adding "irregular grid" to this description.  Perhaps here: "in steps of 250m, to produce an irregular grid with a total number"

We updated this mentioning individually adjustable range bin sizes.

p5 Figure caption. "Example" rather than "exemplary"

Done.

p7 L179. "Extensive use" rather than "Excessive use"

"Excessive" has been dropped.

p7 L180. Replace "lightened" with "simplified"

Done.

p7 Equations 5-8. I think the variable names and subscripts could be chosen to better indicate which set of variables are the raw signals and which are the cross-talk corrected signals. It's particularly confusing that the pair with mixed Rayleigh and Mie components are subscripted "ray" and "mie" while the pair where they are actually separate is generic with no mnemonic. Using p and m subscripts for the corrected measurements in 5 and 6 might help.  I would also suggest renaming the variables in 7 and 8 to avoid the ray and mie subscripts on the raw channels (although I know that suggestion might be more controversial because it's based on historical precedent with this kind of instrument).

We fear that having inconsistent variable naming conventions compared to the main reference (Algorithm Theoretical Baseline Document, Flamant et al. 2017/2020) would further complicate understanding. This way, it is mostly aligned. The only difference is the lowercase s for signals, so to circumvent confusion with the error covariance matrix S_y (capital).

p8 L203 Delete "can be rephrased into".  The slash notation is confusing and unnecessary. In this case "are equivalent to" is the better phrase.

The sentence has been adapted this way.

p8 L209 Appendix, not Annex

Done.

p8 L227, I believe the meaning will be clearer if you remove "to account for" and instead say "because additional noise contributions, such as ..., are likely to dominate over the Poisson noise"

The other contributions are not likely to dominate over Poisson noise, but rather to increase it noticeably and to smear it out in a quasi-Gaussian fashion, so we prefer to keep the statement as it is.

p9 L246, "In theory" instead of "Principally"

Done.

p9 L252, "simplify" instead of "lighten"

Done.

p9-10, L261-263, I had trouble understanding this sentence until I read the Appendix. It might be clearer to move "scaled" and break the sentence into two: "Here we use the variance measured at 2.9 km resolution, scaled to approximate the noise level in the 87 km bins.  This approximation assumes the scene is homogeneous so that all the variability is due to measurement error."

Shorter sentences are always favourable, so we adapted the sentence.

p10 L269 replace "artificially increased" with "increased"

Done.

p10 L272-275. I found these few sentences very confusing. I think the authors are saying the extinction bias due to underfilled bins tends to decrease the measured range of lidar ratio, but that this is not a compelling reason to reduce the lidar ratio upper bound in the algorithm, so they end up ignoring the bias found by Flament et al. 2017. If the effect is ignored, then these sentences are somewhat of a distraction and could simply be deleted. If the authors feel that it is important to keep these sentences about the bias due to underfilled bins, (and if my interpretation is correct), I think the readability could be improved by (1) signalling the contrasting thought using "On the other hand" (or "In contrast") instead of "Additionally", (2) using "aerosol partially filling a range bin" instead of "different hypothesis on the distribution of aerosol layers within a range bin" (3) specifying "underestimate" instead of "alter" and (4) correcting "co-polarized particle backscatter coefficients" to "co-polarized measurements". That is, the bias due to underfilled bins is a bias to extinction, not to particle backscatter coefficients.

We changed the text to: "On the other hand, as shown in Flamant et al. 2017, aerosol partially filling a range bin can easily underestimate the true particle extinction retrieval results by a factor of 16. The applied bounds need to account for these forward model errors by an extra margin."

p10 L288. Forty or forty thousand? Forty seems more likely, in which case three digits after the decimal point for an integer is a strange instance of false precision (should be just "40"). Or if forty thousand, it should be a comma not a point "40,000" - but in that case, 40,000 is a crazily large number of iterations. What is the typical number of iterations actually required for convergence?

When just one profile is considered, as in problem (14), then the number of iterations is on the order of 50 (roughly 0.5 seconds on an office laptop) to achieve convergence (an average cost function value per bin below 1). In practice, however, we are interested in solving problem (15) for all profiles simultaneously. In this case about 5000 iterations are required (roughly 45 seconds on an office laptop) to generate good optical properties products for a whole orbit (roughly 450 profiles). This information has been added to the manuscript.

So essentially, solving (15) for the whole orbit saves us time. Though, on second thought there must be a better converging formulation, but we have had no reference on this. We made an attempt to reformulate our model in terms of log(L_p), but the convergence speed did not increase noticeably. It is very likely that a more advantageous scaling exists since the L-BFGS-B algorithm is not scale invariant. Unfortunately we haven't found it.

p10 L289. "many fewer" instead of "much less"

Done.

p11 L305 Consider inserting Equations C2 here. The sentence could read "This relation is inverted to produce Eq. C2", and deleting the terminology "Moore-Penrose pseudoinverse" in the main text, since it's more understandable in the Appendix where it is explained more completely.

We dropped the phrase regarding the Moore-Penrose pseudoinverse and changed it into a reference to equation C2.

p12 L339. "optically thin" rather than "thin"

Done.

p12 L350. "at the price of" instead of "to the price of"

Corrected.

Figures: I hope Figures 2-4 will be larger with proportionately larger text. They are quite difficult to read.

Done.

P14, L361 "either" not "wither"

Done.

P14, L381 "either" not "wither"

Done.

P14, L383, I'm not sure what "locks extinction and backscatter to appear colocated" means. Does it mean something like "MLE retrieves extinction and backscatter at the same effective resolution"?

We changed it into "forces extinction and backscatter to appear colocated", i.e. to appear together in the same range bin.

Figure 4 caption. "CALIPSO Feature Mask". Better to use the official CALIPSO product name and be specific: is it the "Vertical Feature Mask (VFM)" or the "Atmospheric Volume Descriptor (AVD)"?

Sorry for the sloppy notation, we meant the vertical feature mask (VFM) and corrected the mentionings thereof in the text.

Figure 4 caption, the sentence starting "The rightmost column shows the Calipso Featuremask" is confusing. This could be rearranged to first describe the rows as backscatter, extinction, and lidar ratio. Then in a separate sentence, say the CALIPSO feature mask is repeated on each row.

Done.

Figure 5 caption, "error bounds" rather than "errors"

Done.

P17 L408-409. Delete "and it might be horizontally and vertically". It is disrupted, not might be.

Done.

P18 L419. I suggest using "copolarized" rather than "co-polar", to be consistent with how you have described it earlier.

Done.

P18 L427. Explicitly include copolarized lidar ratio in this sentence: "expected values of copolarized lidar ratio of 80sr - 120sr for desert dust". It would be good to avoid the possibility of these expected values being taken out of context and mistaken for the more usual non-polarized lidar ratio.

Done.

P18 L431. Replace "all values" with "expected values" and replace "the estimated error ranges" with "the error ranges estimated from Eqs C2 and C3" (or some other phrase to make it clear that it's the error ranges calculated from the new methodology. They do not fall within the "poisson" error bounds.)

Done.

P19 L449. ":Hence, " is not the right connector since it implies causality, and consequently I'm not exactly sure I know what is meant. I think the authors are only explaining what was meant in the first part, so perhaps replace with "; that is, ".

Done.

P19 L470. Replace "with this" with "in addition". Again "with this" is usually used idiomatically to indicate a causal relation.

Here we actually mean to indicate a causal relation, because it is only by the a-priori knowledge of constraints in the first place that noise can be detected and suppressed in a second step.

P19 L472-474. The flow of this paragraph is interrupted. I suggest moving these sentences about how the coupled retrieval improves backscatter to the earlier spot at L469 immediately after the statement that backscatter is improved along with extinction. The information about the constraints and the anti-correlated noise are separate thoughts.

We agree and moved the sentence on the reasons for the improvements to the end of the paragraph.

P20 L483. "suitable" rather than "suited"

Done.

P21 L499. "the properties above" doesn't make sense here, since no properties have been defined yet in the Appendix.

Has been changed to "the properties in section 3".

P21 L515. Keep "are equivalent to" and delete "can be rephrased into"

Done.

P21 L521, I suggest replacing "optical depth/extinction" with "layer optical depth". Dropping "extinction from this sentence is no loss since the relationship with extinction is described in the very next sentence.

Done.

---

## Author Response (AR2)

*Dear Editor and Reviewer,*
*We have carefully read your feedback and answered in much detail, whenever necessary for the matter. The questions and remarks were very helpful to refine our text and we have done our best to account for all remarks. With our most recent edits we hope to have clearly underlined the most important aspects of the new method, because some points cannot be simplified further than this, contrary to what has previously been suggested by the reviewer. This regards the 'single root cause' of the improvements, which is due to an interplay of (i) consideration of measurement errors, (ii) coupled retrieval of all state space variables and (iii) the implementation of constraints. No single aspect can be regarded as the root cause for the improvements, because all of them have evident influence on the refined solution, as we explain below and have now also made clearer in the manuscript.*
*We have also gladly adopted most of your suggestions regarding the calculation and display of error, except for mainly Figure 8, where time and lack of methodology did not allow for error bars in the MLE case. However, a reference to the simulation case has been added for a qualitative assessment in this case.*

Line numbers in the following comments refer to the "Tracked changes" version of the document, amt-2021-212-ATC1.pdf.

Major Theme 1: there are various stated reasons for why the current method is better than the SCA; it would be good to emphasize if there is just one that is the key reason (which I think there is), and downplay those that are more minor.

*This is a good and helpful remark. As you will see in the following answers, there are indeed three reasons for better performance: i) consideration of uncertainty, ii) coupled retrieval and iii) the constraints. Of these, reason ii) and iii) dominate in our opinion, while i) is still a rather important aspect. Also, if one left out constraints in a coupled retrieval or constrained a decoupled retrieval, not much impact would be expected from our side. This was highlighted in the text, see l. 80-90.*

First I want to say, in the paragraph at L250-256, thanks to the authors for adding this note about how the MLE and SCA solutions are algebraically the same when the box constraint is removed from MLE and the zero-flooring is removed from SCA. This insight was very helpful for my understanding of both retrievals, and makes the paper better. But now this better understanding has caused me to question several other details, and I think it's possible to have a more consistent picture throughout the manuscript of how the methods differ and what the impacts are.

The various reasons are first introduced at the paragraph at L76-91. (1) "we account for the noise of the signals" (2) the new retrieval is simultaneous with respect to backscatter and extinction (3) constraints are applied (which are applied differently than the zero-flooring of the SCA) and (4) "dominant anti-correlated noise that originates from the cross-talk correction" is "automatically detect[ed] and suppress[ed]".
*We clarified the text by rephrasing (4), but reason in a later comment that (1)-(3) are all important aspects, see l. 80-90.*

Reading between the lines of the manuscript, it seems the SCA implements its zero-flooring constraint after the mathematical retrieval of backscatter and extinction; …

*For clarification we added a sentence close to line 225, mentioning that the zero-flooring in SCA happens indeed during the iterative retrieval (not just afterwards), see Flamat et al. for reference, and that SCA midbin is not being regularized in this way.*

...does this mean it breaks the connection with the measurements, such that the final reported backscatter and extinction solution cannot reproduce the measurements well? If that's correct, I think that's probably the key difference, since the new retrieval, in contrast, implements the box constraints as part of the retrieval and requires the retrieved backscatter and extinction to be consistent with the measurements subject to the constraints. So, it's a better retrieval because it is self consistent and preserves consistency with the measurements. Do the authors agree? …

*Indeed, we expect as well that the SCA solution with zero-flooring does not reproduce the measured molecular signal perfectly (see https://doi.org/10.5194/amt-2021-181, Figure 7), in contrast to the MLE case. On the other hand, this should not be the case for the SCA midbin (SCA MB) results, which are not regularized ad-hoc and hence correspond directly to the measured signals.*

...If so, this is basically the same as the reason marked (3) above, but I believe it should be possible to clarify the introduction, discussion and conclusions to make it more obvious. I believe it would also be helpful to remove or downplay the other distracting reasons that are less informative, or to relate them to this main reason.

*In order to highlight this misfit of the SCA solution wie added close to l. 260: "Due to the implemented zero-flooring of optical depth in the SCA, its retrieved optical properties do not correspond perfectly to the measured signals, though $J\_obs=0$ still holds in the case of the unregularized SCA MB retrieval." The important aspect here is that the MLE does still perform better than SCA MB, which is not regularized.*

As for the other three reasons: In backwards order, first the "dominant anti-correlated noise". In my first review, I had trouble interpreting the impact of the anti-correlated noise and the authors response says this is actually not important. In that case, I think this reason should be dropped from L84.

*We did so by rephrasing into "A considerable gain in the quality of the retrievals is expected [...], because a coupled retrieval in conjunction with a box-constrained set of space variables will allow for important information exchange during the determination of the self-consistent set of optical properties."*

At L 520 "With this [the introduction of constraints] the anti-correlated noise in the cross-talk corrected signals can be traced back and effectively suppressed", I offer an alternate version that I think is more in line with the author response and revisions in the Appendix: "Since the box constraints are integral to the simultaneous retrieval of backscatter and extinction, noise is suppressed in both products simultaneously, in contrast to the zero-flooring of the SCA

which works on the channels independently without regard to the fact that the errors are correlated due to channel cross talk (Appendix A)."

*We appreciate your effort and adopted your suggestion with minor changes, namely: "Since the box constraints are integral to the simultaneous retrieval of backscatter and extinction, noise is suppressed in both products simultaneously, in contrast to the zero-flooring of the SCA, which considers the extinction variable independently without regard to the fact that the errors are correlated due to channel cross-talk (Appendix A). Furthermore, the results of SCA and SCA MB do not strictly fall into a physically meaningful subset of solutions, e.g., they include negative backscatter coefficients."*

Next, the simultaneous retrieval of backscatter and extinction. I think this is important too, but not in the way I first interpreted the writing. L 516 says "A coupled retrieval may improve the precision". However, the authors have now convinced me that if it weren't for the constraints (box constraints for MLE and zero-flooring for SCA), both retrievals are the same algebraic noise-fitting solution, since they have the same number of measurements as unknowns. So, in the absense of the constraints, it doesn't matter at all if the backscatter and extinction are retrieved sequentially or coupled. Rather, as the authors emphasize, it's the better implementation of constraints, an implementation that minimizes disagreement with the measurements, that improves the precision. It's much easier to implement these better constraints in a simultaneous retrieval, of course. So, I think it's something like "A retrieval that implements constraints simultaneously with the retrieval of backscatter and extinction improves precision".

*We fully agree to the point, that the interplay between coupling and constraints is the root cause for precision improvements, as you figured out correctly. One without the other will not have the desired effect. We therefore clarified this in the text as well, see l. 510 to 530.*

Finally, does accounting for the noise in the signals improve the retrieval? As the authors have pointed out in the discussion and revisions, there are the same number of unknowns as measurements, and so the solution (in the absense of the box constraint) is just the algebraic solution. In that case, it doesn't matter what the measurement error covariance matrix looks like; the same solution will be found. What about with the box constraints? No, I think with no prior or regularization term, I believe the minimum cost is still independent of the measurement error covariance matrix, although that minimum won't be zero. In the implementation of the retrieval, if the iterations are cut off at some thereshold that's dependent on the measurement error covariance matrix, then of course that matrix would impact the solution in that way, but if the algorithm actually converges to the minimum cost function (which I believe is the intent here) then I think the measurement error covariance matrix will not impact the solution. Am I correct?

*This is not correct, because it is simplified too far. The entries of the covariance matrix have an influence on the solution in the constrained case (but not in the unconstrained case). Consider the following toy model:*

*Let's model something like the ratio of received molecular signal S_meas to the expected (simulated) molecular signal S_sim and let's call the ratio S_meas/S_sim = X. Now, let's also assume we measure this ratio (directly, i.e. crosstalk is known) at two different distances r_1*

< r_2. Then, we also know that due to additional aerosol attenuation $X(r_1) \geq X(r_2)$ must hold, which is our constraint. Shorthand, we just note $X_i = X(r_i)$. Now, assume a measurement noise covariance matrix on X, which is diagonal (uncorrelated measurements) with variances $V_1$, $V_2$. We call $X_i$ the state space variable and $Y_i$ its measurement $Y_i = X_i + epsilon$ with noise term epsilon.
The cost function then is $J = 1/V_1 * (X_1-Y_1)^{**}2 + 1/V_2 * (X_2-Y_2)^{**}2$.

Now there are essentially two different cases / scenarios
i) $Y_1 > Y_2$
Assume e.g. $Y_1 = 1$ and $Y_2 = 0.9$. In this case, we can minimize the cost function with $X_1 = Y_1$ and $X_2 = Y_2$, which is in line with the constraint $X_1 > X_2$. Here, the variances do not contribute to the solution and the cost at the solution is J=0.

ii). $Y_1 < Y_2$
Assume e.g. $Y_1 = 0.9$ and $Y_2 = 0.95$. In this case, the solution $X_1 = Y_1$ and $X_2 = Y_2$ does not fulfill our constraint of monotonically decreasing signal amplitude and can therefore not be chosen.
What classical SCA does (in principle) is to say: Let's set $X_1 = Y_1$ in the first place, and since $X_2$ cannot be greater than $X_1$, we make the best guess of "no attenuation" between the points and assume $X_2 = X_1$.
This option corresponds to a cost of $J = 1/V_2 * 0.05^2 = 0.0025/V_2$

Now, the MLE considers that both values $Y_1$ and $Y_2$ are uncertain. If $V_1 = V_2 = V$, then the state space and the corresponding cost function would look like in this figure (note that "half" of the state space is cut away due to the constraint):

[Figure]

Here, the red dot marks the minimal cost. Which suggests $X_1 = X_2 = 0.925$ as global minimum. By setting $X_1 = X_2$ one can reproduce this solution (minimize J for $X_1$ after substituting) and see that it corresponds to the weighted mean of the measurements $Y_1$ and $Y_2$, which becomes the arithmetic mean when $V_1$ is equal $V_2$:
$X_1 = X_2 = (Y_1/V_1 + Y_2/V_2) / (1/V_1 + 1/V_2) = 0.925$
Not only is the remaining cost at this solution lower than in the SCA guess, namely
$J = 1/V_1*0.025^2 + 1/V_2*0.025^2 = 2/V*0.025^2 = 0.00125/V$,
but we also see that the solution explicitly depends on the entries of the error covariance matrix (not their absolute but their ratio). If e.g. $V_1 = 4*V_2$, then $Y_2$ would have higher

*weight in the solution, which became  X_1 = X_2 = 0.94, see also illustrated in the figure below:*

[Figure]

*This toy model generalises immediately to the problem at hand: The positivity constraint on optical depth is equivalent to a monotonically decreasing molecular signal ratio X with increasing distance from the satellite. The measured signal ratio X is noisy, and so, case i) and case ii) take almost equal share. Especially when the range bin thickness is reduced, the noise (and its estimate) will increase and the MLE solution at the interface of bins with different width will give higher weight to the thicker bin. We hope this clarifies that the relative magnitude of the covariance matrix entries will have significant influence on the solution in the constrained case, whereas the absolute scale of the uncertainty solely determines the planned convergence criterion, which optimally does not impact the solution.*
*We made a remark in line 292: "Note that the position of a cost function minimum is invariant to scaling of the covariance matrix $\mathbf{S}_y \to \lambda\mathbf{S}_y$ with scalar $\lambda$, while the relative magnitude of its entries is important when being subject to constraints, i.e., in a general case where $J>0$ holds at the minimum."*

If so, then almost everywhere that the measurement error covariance matrix is mentioned is confusing and somewhat spurious, and should probably be reconsidered. E.g. Line 80 (mentioned above, where better performance is attributed to accounting for signal error); and L 273 - I understand of course the desire to avoid the complication of the off-diagonal terms in the covariance matrix, but if the covariance matrix doesn't impact the solutions, perhaps this isn't very relevant; and L 279 - "As pointed out by Povey et al (2014) unbiased estimates are a prerequisite" - in Povey et al, it's a prerequisite for an optimal estimation retrieval that has a prior term that must balance with the measurement term, but this retreival does not have that feature.

*As demonstrated above, the choice of the error covariance matrix will influence the solution in presence of constraints.*

Major Theme 2: There's a need for better characterization of the error distribution of the box-constrained MLE retrieval results. The authors have shown that the new retrieval produces a better-looking solution than the SCA and SCA MB retrievals, but they also have shown that it is still quite impacted by measurement noise and the resulting error is quite significant (100% or more). For that reason, the results should not be used without an understanding of their uncertainty. The simulation cases, therefore, are a hugely important part of this manuscript. It is also important to include some kind of estimate of the spread of

solutions for the included real data cases as well. Here are the easiest ways I can think of to do this.

Figure 7. Each solution appears to be an average of multiple bins. Can the figure include an indicator of the actual spread of solutions obtained from each averaging interval?

*This is a very neat idea, we therefore display now the range of maximum and minimum lidar ratio values within the bins used for each data point and updated the caption. This gives the reader a clear idea of the retrieval method's behaviour without hypothesising on the signal noise.*

Figure 8. I appreciate that this case is harder, since it is only two individual profiles. One approach is to produce another simulation with profiles taken from the solution of this real data case (i.e. same backscatter and extinction as one of the profiles or the mean of them). If it is difficult to use the instrument simulator to do this, then I think it would be accpetable to use the assumed measurement error covariance matrix (i.e. from Eq 13) to estimate it. (This is the clarification of my earlier suggestion that the authors requested.)

*We agreed to delete the appendix on analytical error propagation, as it is not used throughout the manuscript anymore. Therefore, we would want to mitigate confusion by relying on the non-representative, analytic way of calculating uncertainties here, without having it documented anymore. Due to the time constraints, we were not able to find an alternative for the error propagation, so we have to keep Figure 8 without error bars on the MLE results. But we suggest to add the following text within section 4.4 for a qualitative error assessment:*
*"By comparison to the similar simulation case I, we expect a backscatter coefficient error in MLE retrieval on the order of 30 \% within the aerosol layer below 3.5 km and an extinction coefficient error on the order of 100 \% for individual range bins. A comparison of SCA and SCA MB results with the simulation case I and the ground truth also suggests, that the currently reported error in the L2A product is no reliable estimate."*

Ultimately, the authors would like an analytical error calculation since only an analytical solution is believed to be fast enough for practical use in real data processing. However, since both the input errors and the forward model (when box constraints are included) are non-linear, any analytic solution with assumptions will be hard to accept until sufficient research and analysis demonstrate consistency with existing numerical solutions for real data. So, a fair number of numerical solutions will be required to be calculated anyway. This underscores my hope that it's not unreasonable to wish for numerical results for the specific cases in the manuscript.

*We agree that any analysis of an analytical approach will need to rely on a manifold of simulation cases. But for this comparison to be made, we would need to develop an analytic model / an alternative approach in the first place. You also acknowledge that such a comparison will take considerable effort. Hence, we will have to work on the error characterisation in the future work but cannot include it within the rather limited time frame of this revision.*

I would also suggest completely removing the analytic error propagation. E.g. the paragraph at lines 324-334 and Appendix C. The authors know it does not correctly represent the error propagation of their retrieval because it doesn't include the box constraint. For that reason, they have deleted all the results relating to this error propgation. So, it should not still be included in the methodology. I think everything after "whereas" at line 325 should be deleted (as well as Appendix C). If desired, the authors can make a short statement acknowledging that an analytical error propagation for the current retrieval would have to include a way of representing the impact of the box constraint, which is a topic of future work.

*Given the earlier discussions, this is a reasonable measure. We shortened the paragraph at lines 324-334 significantly and added a note regarding error propagation from linearisation: "While a similar analysis can be made for the MLE, we find that especially the obtained extinction uncertainties would strongly overestimate the actual variability of MLE results due to the omitted constraints in such procedure. Hence, future work regarding the implementation of the box-constraints in the error estimation is pending."*

*We could potentially keep Appendix C for further reading, but it is indeed obsolete with respect to the results. So we decided to remove it as suggested, together with all references thereof.*

Minor Theme 1, related to Major Theme 2. The representation of the errors for the simulation cases can be improved.

Figure 2 and Figure 4 make clear that the standard deviation is not a good representation of the spread of the errors in the box-constrained MLE method (so perhaps also for the SCA and SCA MB as well), in that the shaded area for the lidar ratio goes well below 2 sr, although the constraint makes it impossible that any solutions lie in that range. I encourage the authors to remake the figure with a different visualization of the spread that is more representative of the actual range of the results. An alternative is to use a percentile spread, such as the 25-75% limits. (In figure 3, since standard deviation is shown along with a bias statistic, I think it's more acceptable).

*The primary goal of the visualization in Figures 2 and Figures 4 is to compare the methods. For your request I replotted the data in these figures using a 25 and 75 percentile. Doing so, another problem occurs in the SCA case (see figure below): Since the majority of extinction values below the first bin are zero, there is often no visible spread as both the 25 and 75 percentile are also zero in mid latitudes. Unless we take higher percentiles, this problem cannot be solved, but too high percentiles are not representative of the bulk data anymore. Although the distributions for SCA and MLE are non-symmetric due to the constraints, we suggest keeping the standard deviation as an established measure of spread (and error) for the sake of comparability of the retrieval methods and with Figures 3 and 5. We appreciate that penetrating towards the negative x-axis is unnecessarily confusing for the MLE case and therefore clipped the error bars at zero for the MLE lidar ratios. We hope to have explained this point well and that this is a satisfactory compromise, as we believe that we cannot find a statistical measure of spread that perfectly suits all different empirical distributions for SCA, SCA MB and MLE variables.*

[Figure]

[Figure]

Previously I asked about the potential for bias caused by the box constraint algorithm and the characterization of the error output to account for potential bias. The authors have added a line late in the paper that suggests that an observed bias may be due to the box constraint, and I appreciate that, but they also admitted they do not have a good understanding of the algorithm. I think more analysis is required to understand how the algorithm affects the distribution of solutions. I suggest one way to gain a better understanding is to show a histogram of the solutions from the synthetic cases, particularly a histogram of the lidar ratio with mean and median marked. For instance, it would be good to see the behavior near the edges (e.g. 2 sr), whether the histogram looks truncated or rather "piled up".

*The reason why we do not provide the LR statistics is due to the fact that the LR histogram is not representative of the results. That is because, when the algorithm retrieves optical depth close to zero, the lidar ratio can be arbitrary (and indeed bunches at the first guess and the borders, etc.). In order not to confuse the reader, we report the lidar ratio defined from the mean backscatter and mean extinction, effectively throwing out the influence of "(almost) empty" bins with diverging error. Therefore, it is true that the lidar ratio is directly retrieved. But in contrast, backscatter and extinction are always well defined properties, which is why they are preferred. So, showing a lidar ratio histogram would mislead the reader, as all samples would be assumed to have equal weight.*

Figure 3. I like the new Figure 3 very much; it is very helpful to see the behavior specifically in the region with significant aerosol, and helpful to see a bias calculation paired with the standard deviation, and to have equations at line 364-365 specifying the statistics. I would

also like to see panels showing the statistics for lidar ratio (which is, after all, a directly retrieved quantity from the MLE retrieval.)

*First of all, thanks for the positive feedback on the improved visuals. The reason we did not include a seperate panel on lidar ratio is to be economical in terms of space. Mean and error of lidar ratio appear clearer in Figures 2 and 4 than for extinction and backscatter due to their linear scale and the piecewise constant ground truth.*

Figure 5. Figure 5 should have the addition of the zoomed in boxes, like Figure 3 has.

*Thanks for the remark, but considering the wide spread of the results on the x-axis, the visualisation does not considerably improve by zooming in solely on the y-coordinate. For Figure 3, this was different, as there was considerable "bunching" in both directions close to the ground. We therefore suggest saving up this space.*

Minor theme 2. Items related to the discussion of ratioing of signals and discontinuities where the range bin size change.

L 367-370 (a). First, I suggest replacing "seems to be triggered by the refined range bin" with a more definite description of the observation, holding the hypotheses for the next sentence. That is, "the bias is colocated with the change in range bin size". I suggest this because I think the change in range bin size only makes the problem more obvious, but does not actually cause the problem (more below).

*We included this small change.*

L 367-370 (b). Next, about the first theory about the bins that are not uniformly filled: does this make sense? Wouldn't the requirement for uniformly filled bins be more badly violated by large bins and better met by small bins? If the suggestion is that it is worse here due to the dramatically increasing slope of the aerosol extinction profile, then the bias would logically be colocated with the slope but only coincidentally colocated with the change in bin size. Flamant et al. is quoted elsewhere as predicting a bias in extinction due to this reason, but is it also expected to affect backscatter, as here? If this part is kept, Flamant et al. should be referenced here, with a specific description of their work showing how non-uniformly filled bins cause a bias.

*We made this point referring to the gradient in the ground truth profile, which is steepest (coincidentally) in the region with the lowest bin size. On second thought, I would argue that with this gradient direction, one would expect underestimated backscatter, since the molecular attenuation happens mostly in the lower part of the bin (increasing X), while the aerosol backscatter Y remains roughly similar. Hence, the backscatter coefficient beta ~ Y/X would need to decrease in this area by applying the hypothesis of uniform bins.*
*Therefor, we deleted this sentence.*

L 367-370 (c). Finally, I find the other theory more convincing (ratioing of noisy signals). But why doesn't it apply to the MLE retrieval as well?

*A glimpse to the reason was provided by our statement "..., because here mean(beta) will become biased high increasingly with increasing uncertainty of Xi." Now, we completed the argument as follows: With MLE, we constrain the possible values for X_i to a physical subset, which makes them effectively less uncertain compared to SCA and SCA MB. Hence, a lower bias is obtained. This was added to the text.*

L412-413, "due to its noise suppression capabilities" is presumably the answer to my question but I find it somewhat vague. Can it be made more specific?

*With the addition to the prior subsection (see answer above), we suggest to keep this statement as is and hope to have clarified this point.*

L579. "if this varying reliability of the signals is not taken into account". I believe from L367-370 that the bias is linked to taking the inverse of a noisy signal, not because of a discontinuity in bin-size, although it is more noticeable because the discontinuity in bin-size results in a discontinuity in the bias.

*Thanks for stumbling over this statement. It is our fault we did not make clear that this paragraph is dedicated to the extinction variable. We completed this argument as follows: "Hence, if this varying reliability of the signals is not taken into account, biases or oscillations in the extinction variable can potentially be triggered by zero-flooring whenever range bin heights change. This is due to the fact that extinction essentially depends on the moving ratio of noisy signal values along an atmospheric column. The mean absolute of this ratio increases with increasing noise, which may then lead to a bias after zero-flooring negative values in SCA, see also (reference to https://doi.org/10.5194/amt-2021-181) for a graphical explanation and a showcase without flooring."*

Miscellaneous minor suggestions in line-number order.

L 37. Illingworth et al. 2015 is a secondary reference with respect to the idea that the indirect effect of aerosol on clouds is the largest uncertainty in radiative forcing. I suggest referecing some primary sources, or a major climate change review such as the IPCC report.

*We now reference the IPCC report directly.*

L303. "not invariant under variable transforms". While admittedly I only quickly skimmed Zhu et al (1997), it caught my eye that they say the algorithm is indeed invariant under transforms with the exception of the first step away from the first guess. This suggests to me that efforts to find a better first guess might be better rewarded than attempts to find a better transform to deal with a poor first guess. The aerosol-free atmosphere is a difficult first guess to work with. The standard HSRL technique provides backscatter in one algebraic step. Perhaps consider calculating backscatter from the ratio of the channels, and with this estimate optical depth using your first guess lidar ratio of 60. (I am not suggesting this is required as a response to this review but offer the suggestion in case it's helpful.)

*Just above the lines you are referring to, Zhu et al. state in section 4.1: "However, complete scale-invariance was not possible to achieve; indeed the limited-memory algorithm itself is*

*not invariant to linear transformations in the variables." This is also what we observe. The statement you refer to is "However, the algorithm is invariant with respect to scalar multiples of the variables and the objective function, and we have been able to maintain that invariance in the code with only a few exceptions." We interpret these statements such that J -> lambda\*J and v -> lambda\*v with scalar lambda, cost J and variable vector v are symmetries of the algorithm. However, the algorithm is not invariant to the general case v -> M\*v with a matrix M, as the first statement about linear transforms implies. Not being invariant to linear variable transformations implies that there is no general invariance to nonlinear variable changes (e.g. taking the logarithm) either.*

L312. I'm curious about the description of running 40000 iterations so that "the estimate should fit as close as possible to the signal data". Does the cost function really continue to decrease for 40000 iterations? I believe it's common for the cost function to begin to jump around after a while and not continuously decrease. I agree with the point that cutting off the iteration prematurely leaves some unnecessary impact from the first guess (especially if the measurement error covariance matrix is not strictly correct), but I do think it should be cut off when the cost function ceases to decrease.

*In our current implementation it continues to decrease, but with very low speed. That is why we checked additionally with the total cost criterion, in order to assess at which point no additional information can be extracted from the measurements (whenever the modelled signal values are on average maximum a standard deviation away from the real signal).*

L 378. "the feedback" is vague. Does it mean that when the SNR decreases there is a greater proportion of negative solutions that get filtered out by the zero-flooring, and therefore bias the mean solution?

*This is a reference to the topic of line 579 (see comment above). We explain this feedback in the Appendix A as written above and added a reference to line 387 for the interested reader.*
L380. At the start of the added section, it would be good to say "with the exception of the bin closest to the surface". This is mentioned in the following paragraph, but the first paragraph is confusing with this omission.

*We fully agree and inserted the remark.*

L 469. Copolarized lidar ratios of 80sr to 120 sr for depolarizing desert dust are attributed to Wandinger et al. 2015. Does that paper really present copolarized lidar ratios? Or is this a calculation of the authors' based on non-polarized lidar ratios from that Wandinger et al. 2015?

*Wandinger et al. explicitly present the discrepancy in Aeolus observations for different types of aerosols, the provided estimates are extracted information from the plots.*

L 415-421. What does the averaging kernal look like below the cloud in the region that has up to 100% bias and up to 500% relative error, but where the average lidar ratio "remains quite accurate"? It would boost confidence in the conclusion, if the averaging kernal also shows that the optical depth is not reliable but the lidar ratio is.

*We are not sure if a look at the averaging kernel can help here, because the high-bias is mainly a non-linear effect (ratioing of noisy signals). We believe this might be a misunderstanding of the terms accuracy and precision? The precision of individual lidar ratio estimates is of course very low, indeed. The presented lidar ratio is obtained from the average backscatter and average extinction over 1000 realisations which have high individual errors. Not the individual lidar ratio is accurate, but this average. To motivate this we added: "This suggests that the noise induced biases in extinction and backscatter do almost balance, which can be motivated by the fact that both variables depend on $X\_i^{-1}$ (the aerosol optical depth equals the normalized log-derivative of pure molecular signal and can be rewritten in terms of the ratio $X_{i+1}X_i^{-1}$)."*

L 517. Geometric overlap? Is that relevant for a satellite lidar like AEOLUS? Is this sentence meant as a more general discussion that also encompasses ground-based lidar?

*See below.*

L 517. I can't see the cross-talk calibration as part of any tradeoff between a coupled or sequential retrieval, since errors in the cross-talk calibration will significantly impact either style of retrieval.

*This sentence was included to highlight why processing schemes for ground-based lidars such as developed by Marais et al. do prefer not to implement a simultaneous retrieval (as stated in lines 87-89). For the specific case of Aeolus it is superfluous, which is why we removed the statement in the manuscript's conclusion.*

L 522 (approximately). It should be repeated in the conclusions that in the simulations moderate amounts of aerosol still cannot be distinguished from zero (despite the fact that the new retrieval does significantly better than the existing one). This can be part of the motivation for the future work with signal accumulation. (And by the way, the manuscript has quite a good explanation of the motivation for the scene-based retrieval strategy.)

*Right, the moderate aerosol amounts cannot be distinguished from zero on a single bin basis, unless one averages over bins, because then the uncertainty decays with approximately 1/sqrt(N).*

*We included: "It is important to note that despite the improvements, moderate backscatter coefficients of about 0.1 Mm$^{-1}$sr$^{-1}$ can still not be distinguished from zero on a single bin basis. Higher precision can only be achieved by signal accumulation or averaging of the backscatter coefficient estimates."*

L 594. "the contribution from the Mie channel in the particle-free atmosphere is pure noise". But C1=C4=1, so the molecular backscatter is distributed evenly across the two channels, so I don't think this is true. Perhpas "the signal in the Mie channel in the particle-free atmosphere is more than half noise"

*Entirely correct, what we meant to refer to was the signal Y. We adjusted the statement according to your suggestion.*

Wording suggestions:

L35. I suggest avoiding the awkward parantheses. Specifically, I suggest removing the parentheses around "optical" and simply deleting "(change)"

*Adopted.*

L40. Consider deleting "so-called". "So-called" usually has a connotation that the speaker does not agree with the label or as a way of using an informal-sounding name in a more formal context, neither of which apply here. I know the intent is "what is called" but it's redundant here anyway, so could just be deleted.

*We deleted so-called from the text, whenever it seemed redundant.*

L276-277. The sentence that starts "This accounts" is confusing, and I'm not sure I really understand it. Can the authors please reword this?

*We replaced it with "This overlap is about..."*

L297. Consider changing to "can cause the retrieval results to underestimate the true particle extinction by a factor of 16, and therefore underestimate the lidar ratio".

*Adopted.*

L314. I didn't follow what "even in unfavorable conditions" refers to.

*This sentence can as well be removed, so we did so.*

L386-387. I suggest deleting "likely due to the diminishing influence of the lowermost optical depth on the cost function". The non-sensitivity of the cost function at this point does not determine that it will over or underestimate, but it does show that a big error was expected, and it's not a speculation. The second part of the sentence can stand on its own without the "likely" part.

*This is right, we rephrased it into saying, that deviations from the ground truth were expected.*

L394. Not everywhere but nearly everywhere.

*Adopted.*

Figure 6 labels and caption. "Lidar ratio" should be "copolarized lidar ratio" (everywhere, but particularly important where measured data is described).

*All figures presenting measurment data have been updated with the term "co-polarized".*

Figure 7 caption. The sentence beginning "The upper error bound" is no longer relevant.

*This was deleted.*

L505. The sentence "It should be stressed...lower error margins" is no longer relevant.

*Deleted as well.*

L579. Change "if" to "since"

*Adopted.*